# High-resolution downscaled CMIP6 drought projections for Australia

Rohan Eccles[1], Ralph Trancoso[1,2], Jozef Syktus[2], Sarah Chapman[1], Nathan Toombs[1], Hong Zhang[1], Shaoxiu Ma[1], Ryan McGloin[1]

[1]Climate Projections and Services, Department of Energy and Climate, Queensland Government, Brisbane, Australia
[2]School of The Environment, The University of Queensland, Brisbane, Australia

*Correspondence to*: Rohan Eccles (rohan.eccles@gmail.com)

**Abstract.** Climate change is projected to lead to changes in rainfall patterns, which, when coupled with increasing evapotranspiration, has the potential to exacerbate future droughts. This study investigates the impacts of climate change on meteorological droughts in Australia using downscaled high-resolution CMIP6 climate models under three Shared
Socioeconomic Pathway (SSP) scenarios. The Standardised Precipitation Index (SPI) and the Standardised Precipitation Evapotranspiration Index (SPEI) were used to assess changes to the frequency, duration, percent time, and spatial extent of droughts. There were consistent increases in droughts projected for south-west Western Australia, southern Victoria, southern South Australia, and western Tasmania using SPI and SPEI. There were significantly larger increases for SPEI derived droughts, with consistent increases projected for most of the country. Increases to drought appear to have mostly come at the
expense of 'normal' climatic conditions, with similar or increased time spent under extreme wet conditions, indicating an overall shift towards more extreme climatic conditions. The largest increases occurred at the end of the century and under the high emissions scenario (SSP370), demonstrating the influence of emissions on extreme droughts. For instance, if emissions reached high levels by the end of the century, the area subject to extreme drought in drought prone Southern Australia would be 2.8 greater than if they were kept to low levels using SPI, and 4 times greater if assessed using SPEI. The insights generated
from these results and supplementary tailored datasets for Australian Local Government Areas and River Basins are essential to better inform decision making and future adaptation strategies at national, regional, and local scales.

## 1 Introduction

Droughts are among the costliest climate hazards in the world, with significant ramifications for agriculture, society, and the environment (Cook et al., 2018). Between 1998 and 2017, droughts were estimated to have cost $2.3 trillion dollars (USD), affecting 1.5 billion people globally (United Nations, 2018). Notable recent major drought events have occurred in California (He et al., 2017), the Mediterranean (Kelley et al., 2015), and in Australia (Van Dijk et al., 2013). The recent Australian Millenium drought which lasted from 2001 to 2009 (Van Dijk et al., 2013) was estimated to have cost as much as 1.6 % of the

nation's gross domestic product (Horridge et al., 2005). Compared to other countries of similar population, Australia is disproportionately impacted by drought; ranked 5[th] for economic impacts of droughts and 15[th] for the number of people affected between 1990 and 2014 (González Tánago et al., 2016). A number of studies have highlighted the importance of droughts in Australia, with consequences for a range of other factors including bushfires (Devanand et al., 2024), agriculture (Xiang et al., 2023), water supply (Maier et al., 2013), dust storms (Leys et al., 2023), and public health (Johnston et al., 2011).

In comparison to other natural hazards, determining the onset and severity of a drought event is complex since they are characterised by a gradual build-up, where the largest impacts typically only emerge after many months or years (Kiem et al., 2016). The definition of drought varies according to its application, but can generally be split into meteorological, hydrological, and agricultural droughts (Zargar et al., 2011). Meteorological droughts relate to prolonged deficits in rainfall but may be exacerbated through high temperatures and evaporation, hydrological droughts describe impacts on streamflow and other water

systems (e.g., reservoirs or lakes) (Van Dijk et al., 2013), while agricultural drought primarily focus on soil moisture content (Zargar et al., 2011).

Droughts are usually monitored and assessed through indicators and indices (Svoboda and Fuchs, 2016). Two of the most commonly applied indices for meteorological droughts are the Standardised Precipitation Index (SPI; McKee et al., 1993) and the Standardised Precipitation Evapotranspiration Index (SPEI; Vicente-Serrano et al., 2010). SPI is a rainfall-based index

derived from accumulated monthly rainfall values and can be used to describe droughts at a range of timescales and across different locations. When assessed at shorter timescales (~3 months), SPI has been shown to relate closely to soil moisture and agricultural droughts, while at longer timescales SPI (>12 months) it is more closely related to hydrological droughts (e.g., reservoirs and streams) (Zargar et al., 2011). SPEI is an extension of SPI, calculated as the difference between precipitation and potential evapotranspiration (P – PET), and as such better reflects changes to the overall water deficit by considering the

impacts of both the atmospheric supply and evaporative demand on the water budget. SPEI has also been shown to be more closely related to agricultural impacts than SPI (Labudová et al., 2017; Xiang et al., 2023). The main advantage of the SPI and SPEI over other drought indices is that they provide multi-scalar results that are directly comparable across different regions and climate zones (e.g., arid vs humid regions).

Under climate change, there is potential for more frequent and severe drought events as a result of temperature increases and

changed precipitation patterns, particularly in already drought prone regions (Huang et al., 2016; Zhao and Dai, 2015). Several studies have evaluated the impacts of climate change on droughts using Global Climate Models (GCMs), which have pointed

towards increased drought risk over the 21$^{st}$ century for many regions, including Australia (Cook et al., 2018, 2020; IPCC, 2021; Spinoni et al., 2020). Increased meteorological droughts have been projected for much of Australia (Ukkola et al., 2020; Vicente-Serrano et al., 2022) despite the uncertainties in precipitation (Trancoso et al., 2024). These studies are, however,

based on GCMs with coarse resolutions (~200 km), which have difficulty representing precipitation patterns over complex terrain (Reder et al., 2020) and as such, are not always suitable to provide reliable information to support adaptation and mitigation policy as well as decision-making at regional scales. Additionally, some studies have been reliant on a limited number of climate models, which can have large inter-model and metric-dependent discrepancies leading to uncertain results (Ukkola et al., 2018). There is therefore a need to consider multiple climate simulations as well as high-resolution models to

account for inter-model uncertainties while simulating regional climate granularity.

In order to better represent small-scale features and processes, Regional Climate Models (RCMs) have been employed for drought projection studies across different regions (Gao et al., 2017; Secci et al., 2021; Spinoni et al., 2018), including for regions within Australia (Herold et al., 2021; Syktus et al., 2020). RCMs have been shown to have improved skill in representing patterns of local precipitation and the impacts of topography, coasts, and land-use changes compared to GCMs

(Boé and Terray, 2014; Chapman et al., 2023; Grose et al., 2019; Tian et al., 2013) and may therefore be better suited to study droughts at regional scales. These models (GCMs and RCMs) are the best physically-based approaches currently available to understand future drought processes, characteristics and impacts.

Several studies have considered the impacts of climate change on droughts across Australia (Kirono et al., 2011, 2020; Kirono and Kent, 2011; Mpelasoka et al., 2008) or within a sub-section of the continent (Feng et al., 2019; Herold et al., 2021; Shi et

al., 2020). Mpelasoka et al. (2008) estimated that soil-moisture based drought frequency would increase by 20-40% over most of Australia by the 2030s compared to 1975-2004. Similar increases in drought extent were projected for most regions by (Kirono et al., 2011; Kirono and Kent, 2011). More recently, Kirono et al. (2020) applied SPI and Standardised Soil Moisture Index (SSMI) to calculate projected future droughts using an ensemble of 37 raw Coupled Model Intercomparison Project (CMIP5) GCMs. They projected significant increases to drought hazard metrics, except for frequency, with greater increases

for the SSMI compared to SPI. Herold et al. (2021) used SPI derived from 3 months of accumulated rainfall to investigate changes to 1-in-20-year drought events across southeast Australia with an ensemble of four RCMs. They projected these events would occur approximately 1-in-5 years by the end of the century for large parts of southeast Australia. These studies have, however, relied on projections derived from CMIP5 or earlier.

This study expands on the available body of knowledge for future meteorological droughts in Australia, employing an

ensemble of 60 high-resolution dynamically downscaled CMIP6 simulations (15 historical and 45 future simulations). The downscaling was performed using dynamically downscaled using the Conformal Cubic Atmospheric Model (CCAM), and followed the CORDEX experimental protocol. These projections form part of the Queensland Future Climate Science Program (QFCSP) and are available at a 10 km resolution over the Australian continent as the QldFCP-2 data set (Queensland Future Climate Projections 2). The QldFCP-2 simulations were shown to lead to improvements in mean climate over the historical

period, however, the largest improvements were noted for climate extremes, particularly over coastal and mountainous regions

(Chapman et al., 2023). These projections form part of a national strategy for climate projections, contributing to a wider set of downscaled CORDEX compliant projections for Australia as part of the National Partnership for Climate Projections (Grose et al., 2023), which will underpin climate services and adaptation planning nationally. The objectives of this contribution are:

i)    to assess changes in future projected meteorological droughts, including the frequency of occurrence, duration, spatial extent, and percent time in drought estimated using SPI and SPEI;

ii)    to compare changes in droughts between three different emissions pathways, two categories of drought severity, and two drought indices;

iii)    to evaluate how different climatic regions of the Australian continent are projected to experience future droughts under three different emissions pathways and estimate the time of emergence for significant shifts to occur.

## 2 Methodology

### 2.1 Study Area

This study evaluated changes to drought indices for the entire Australian continent, which encompasses a range of climate regions, including equatorial, tropical, sub-tropical, temperate, Mediterranean, and arid regions. We assess drought changes to four Natural Resource Management (NRM) super-clusters for Australia, namely; Eastern Australia, Northern Australia, Rangeland, and Southern Australia, which are grouped based on a combination of climate and biophysical factors (CSIRO and Bureau of Meteorology, 2015) and have been widely adopted within Australia (Chapman et al., 2024; Grose et al., 2020; Kirono et al., 2020; Wasko et al., 2023) for assessing the impacts of climate change (Fig. 1). Details of the dominant climate zones and ecological characteristics within each of these super clusters are presented in Table S1.

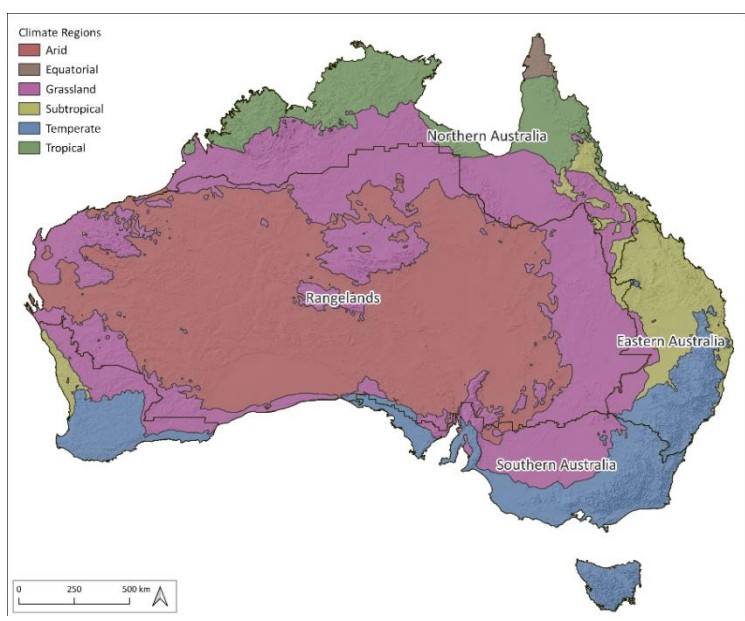

**Figure 1: Extent of study area and sub regions adopted in this study showing NRM super-clusters (CSIRO and Bureau of Meteorology, 2015) for the whole of Australia with major climate regions also shown.**

## 2.2 Data

We used the CCAM model developed by CSIRO (McGregor and Dix, 2008) to dynamically downscale CMIP6 GCMs. Typically, dynamical downscaling involves running an RCM over a limited domain, with the host GCM forcing the lateral boundaries. CCAM differs as it is a global stretched grid model and so is run for the entire globe, with the domain of interest run at a higher resolution. Here, instead of providing lateral boundaries, the regional atmosphere in CCAM is influenced by large scale climate simulated from the host GCM, while at a small scale the atmosphere is allowed to evolve freely (Thatcher

and McGregor, 2009). CCAM was run using a stretched C288 grid in both atmospheric and ocean-coupled versions, which consists of a model resolution of approximately 10 km. In total, 35 vertical layers in the atmosphere and 30 layers in the ocean for the ocean-coupled models were applied (Thatcher et al., 2015). A downscaling approach outlined by Hoffman et al. (2016) was used, which involved bias correcting the sea surface temperatures and sea ice from the host GCMs prior to downscaling. This approach has been found to improve the simulations of climate from CCAM and other regional climate models (Hoffmann

et al., 2016; Kim et al., 2020; Lim et al., 2019).

We used an ensemble of 60 downscaled climate model simulations derived from 11 different CMIP6 GCMs (Table 1). The ensemble consists of 15 runs for historical simulations and three sets of 15 runs for future simulations under three Shared Socioeconomic Pathways (SSP126, SSP245 and SSP370), representing low, moderate, and high-emissions pathways, respectively. The ensemble of GCMs used in this study was selected in order to best represent the future spread in the climate

change signal from the ensemble of global CMIP6 models, while prioritising models which were better able to represent the Australian climate (Trancoso et al., 2023). For instance, we selected several GCMs spread across the distribution of projected temperature and precipitation changes, but also outlier models representing the driest (ACCESS-ESM1.5) and wettest (EC-Earth3) GCMs (Chapman et al., 2023). All the GCMs were assessed based on their ability to represent Australia's precipitation and temperature compared to Australian Gridded Climate Data Project (AGCD; Evans et al., 2020) observational data between

1995 and 2014 using the Kling-Gupta Efficiency (KGE). The climate change signal at the mid and end of the century was evaluated and combined with the KGE score from the historical simulations to select the best performing ensemble runs from the different GCMs through a Skill-Spread-Selection algorithm (Trancoso et al., 2023). Five of the CCAM simulations were run using dynamic atmosphere-ocean coupling as presented in Table 1, in order to better understand the influence of ocean coupling on model outputs. Additionally, three variants including the best performing, the wettest, and the driest ensemble

member from the large ensemble (40 members) of ACCESS-ESM1.5 simulations were considered, to facilitate assessments of intra-model variability. This represents the largest downscaled ensemble of projections in Australia ran at the highest resolution.

**Table 1. Details of the 15 climate model simulations downscaled from 11 CMIP6 GCMs considered in this study.**

| CMIP6 Model | Model full name | Resolution | Ensemble member | CCAM setup |
|---|---|---|---|---|
| ACCESS-ESM1.5 | Australian Community Climate and Earth System Simulator, version 1.5 | 1.875 x 1.25° | r6i1p1f1 r20i1p1f1 r40i1p1f1 | atmospheric atm-ocean coupled atm-ocean coupled |
| ACCESS_CM2 | Australian Community Climate and Earth System Simulator, version 2 | 1.875 x 1.25° | r2i1p1f1 | atm-ocean coupled |
| CMCC-ESM2 | Centro Euro-Mediterraneo sui Cambiamenti Climatici | 0.9 x 1.25° | r1i1p1f1 | atmospheric |
| CNRM-CM6-1-HR | Centre National de Recherches Météorologiques Coupled Global Climate Model, version 6.1, high-resolution | 0.5 x 0.5° | r1i1p1f2 r1i1p1f2 | atmospheric atm-ocean coupled |
| EC-Earth3 | European Community Earth-System Model, version 3 | 0.8 x 0.8° | r1i1p1f1 | atmospheric |
| FGOALS-g3 | Flexible Global Ocean-Atmosphere-Land System Model, grid point version 3 | 2.5 x 2.5 | r4i1p1f1 | atmospheric |
| GFDL-ESM4 | Geophysical Fluid Dynamics Laboratory Earth System Model, version 4 | 1 x 1° | r1i1p1f1 | atmospheric |
| GISS-E2-2-G | Goddard Institute for Space Studies Model E2.2G | 2. x 2.5° | r2i1p1f2 | atmospheric |
| MPI-ESM1-2-LR | Max Planck Institute Earth System Model, version 1.2, low resolution | 1.9 x 1.9 | r9i1p1f1 | atmospheric |
| MRI-ESM2-0 | Meteorological Research Institute Earth System Model, version 2.0 | 1.125 x 1.125° | r1i1p1f1 | atmospheric |
| NorESM2-MM | Norwegian Earth System Model, version 2, 1 degree resolution | 1 x 1° | r1i1p1f1 r1i1p1f1 | atmospheric atm-ocean coupled |

The downscaling approach adopted has been shown to significantly improve the performance over the host GCMs for precipitation and temperature in all seasons when compared to gridded AGCD observational data, with the largest improvements noted for climate extremes, even when assessed across the four Australian IPCC regions (Chapman et al., 2023),

which are similar to the NRM super-clusters adopted in this study. Across Australia as a whole, seasonal precipitation was shown to improve in all models, with an ensemble average improvement of 43% using the Kling-Gupta Efficiency, while the annual cycle of precipitation improved in most models with an ensemble average improvement of 13% (Chapman et al., 2023). Downscaling also improved the fraction of dry days, reducing the bias for too many low-rain days. These improvements have clear beneficial effects for the simulation of future droughts. In the future, the climate change signal of the host GCMs from

downscaling was shown to generally be preserved for precipitation, though with some differences in magnitudes in some regions, particularly in summer. For temperature changes, the downscaled models were shown to have good agreement with the host models across Australia (Chapman et al., 2024).

We used observational data to evaluate the SPI and SPEI indices during the historical period (1980 – 2010). Daily gridded precipitation data with a spatial resolution of 0.05° (approximately 5 km) were obtained from the AGCD. While daily gridded

(resolution of 0.05°) PET data derived from the Penmen Monteith reference crop equation was obtained from the Australian Water Outlook. All observational data was re-gridded to the same grid resolution as the downscaled climate projections using distance weighting interpolation for precipitation and bilinear interpolation for PET.

## 2.3 Drought Indices

We used the SPI and SPEI indices to assess changes to future meteorological droughts based on downscaled climate simulations. SPI reflects changes to precipitation only, while SPEI is calculated from the difference between precipitation and PET and therefore reflects changes to the overall water deficit by considering the impacts of increased temperatures and evaporative demand in addition to atmospheric water supply. To calculate SPEI, we apply PET derived from the Penman-Monteith reference crop method (Allen et al., 1998), which is a physically-based approach. This was calculated offline using

daily CCAM outputs of solar radiation, vapour pressure, maximum and minimum temperature, mean sea level pressure, and wind speed. This method for deriving PET is more intensive than simpler temperature-based approaches but is recommended where data is available (Beguería et al., 2014; Hosseinzadehtalaei et al., 2017; Sheffield et al., 2012).

PET and precipitation data were aggregated to monthly totals for all grid cells and used to calculate SPI and SPEI with the SPEI R Package (Beguería et al., 2017). For SPI, we fitted precipitation data to the gamma distribution, while for SPEI we

fitted the difference between precipitation and PET to the log-logistic distribution as recommended by Vicente-Serrano et al. (2010). Normality tests were performed using the Shapiro-Wilk test at the 95% confidence level on the derived SPI and SPEI to ensure the grid cells conformed to normality. Most grid cells (over 85%) conformed to normality for all months (Fig. S1 in the supplementary materials). As the outputs follow a normal distribution, different categories of drought and also wetness may be classified according to the calculated SPI/SPEI Z-value. Table 2 shows the adopted classification scheme used for both

SPI and SPEI as suggested by McKee et al. (1993).

**Table 2. SPI and SPEI drought classification table following McKee et al. (1993) with associated probability of event from the chosen historical period**

| SPI or SPEI values | Categories | Probability of event |
|---|---|---|
| SPI or SPEI ≤ -2 | Extreme drought | 2.3% |
| -2.0 < SPI or SPEI ≤ -1.5 | Severe drought | 4.4% |
| -1.5 < SPI or SPEI ≤ -1.0 | Moderate drought | 9.2% |
| -1.0 < SPI or SPEI < 1.0 | Near normal | 68.2% |
| 1.0 ≤ SPI or SPEI < 1.5 | Moderate wet | 9.2% |
| 1.5 ≤ SPI or SPEI < 2.0 | Severe wet | 4.4% |
| SPI or SPEI ≥ 2.0 | Extreme wet | 2.3% |

A variety of different accumulation periods may be applied when calculating the SPI/SPEI, ranging from 1 to 48 months. Smaller accumulation periods (1 to 3 months) can be used to assess impacts on systems that are quick to respond to droughts (e.g., soil moisture and small creek flows), while longer accumulation periods (12 to 48 months) better reflect the impacts to slower-responding systems to water deficits, such as groundwater and reservoir levels. We adopted a 12-month accumulation period for our assessments of SPI and SPEI as this was considered as a suitable timeframe for water deficits to impact various

hydrological and agricultural systems (Zargar et al., 2011).

When assessing droughts using historical data, the full period of historical data available is generally used to fit the distribution, with the World Meteorological Organisation recommending a minimum of 30 years (Svoboda et al., 2012). However, when assessing changes to these indices as a result of climate change, a historical period is commonly adopted to fit the distribution. The fitted distribution parameter values are then applied to estimate the SPI and SPEI for the future period, allowing for a

195 comparison of projected future dryness and wetness compared to the recent past. For our assessment, we have adopted a historical period from 1981-2010 to fit the Gamma and Log-Logistic distributions for SPI and SPEI, respectively. Fitted distribution values were then used to calculate SPI and SPEI over the full timeseries, containing both historical and future simulations (1981-2100).

The SPI and SPEI timeseries results are calculated at the grid-cell scale for the observational data and for the ensemble of

200 downscaled climate simulations and are used to detect the occurrence of droughts. For the sake of validation, projected droughts from historical simulations were compared against those estimated from observational data. A drought event is defined when the SPI or SPEI falls below a value of -1 and finishes once the value exceeds -1 again. The definitions for the categories of drought severity are presented in Table 2. In this study, we focus on the changes to all droughts (moderate, severe, and extreme) and to extreme droughts. Metrics relating to the frequency, duration, spatial extent, and percent time in drought

were calculated for each of the drought categories. Here, the frequency is defined as the total number of events recorded over a given time period, the duration is the average duration of recorded drought events (in months), the percent time in drought is the fraction of time droughts occur, and the spatial extent is the number of grid cells affected by each drought severity category divided by total number of grid cells within a given region for each timestep. We evaluated the biases in the drought

metrics from each of the climate models considered compared to the observational data over the period used to fit the distributions (1981-2010).

## 2.4 Climate Change Assessment

We assessed the impacts of climate change on droughts for the 2050s (2041-2060) and the 2090s (2081-2100) relative to the 1995-2014 reference period, which is in line with the IPCC assessment. The historical simulations were used to benchmark the reference period while future simulations were used to quantify the climate change impacts. Results from each of the 45 future simulations were evaluated individually and in a weighted model ensemble, which adopted a one model one vote rule. This weights the models according to the number of downscaled simulations per host model (i.e., the three ACCESS-ESM1-5 models were averaged to a single model, while the two NorESM2-MM and CNRM-CM6-1-HR were also averaged), resulting in an 11-model average. To determine where there is confidence in the changes to the drought metrics, we adopt the signal-to-noise ratio to see where the climate change signal emerges over the 'noise' of the model ensemble (Hawkins et al., 2014). Here, the model uncertainty is considered as noise using the standard deviation of the projections (Hawkins and Sutton, 2011). We calculate the signal from the 11-model average, while the noise is derived from the standard deviation of all 15 projections (Chapman et al., 2024). Stippling is shown on the ensemble mean and median change maps where the signal-to-noise ratio is greater than 1.0 (Chapman et al., 2024; Hawkins et al., 2014; Hawkins and Sutton, 2011).

Results in this paper are also assessed across the four NRM super cluster regions (Fig. 1). Additional supplementary datasets tailoring projected drought impacts to Australian Local Government Areas (566 sub-regions included) and River Basins (219 sub-regions included) are also made available (Eccles, 2024) thanks to the high resolution projections used in this study. We evaluated timeseries results for the individual models and the ensemble average. For this purpose, a 20-year moving average was applied to determine long-term changes to SPI and SPEI values and to remove year-to-year variability. Outputs of both SPI and SPEI follow a normal distribution, with defined probabilities of occurrence for the different drought categories in the historical period (Table 2). We therefore assessed when significant changes to the long-term average values occurred based on a 10% and 20% shift towards dryness compared to the historical period. A 10% shift towards dryness corresponded to the 40th percentile of SPI and SPEI results from the historical period, while a 20% shift corresponded to the 30th percentile. The goal of this analysis was to determine the time of emergence for significant shifts in the long-term climate to take place and to compare the results across regions and emissions scenarios. We also evaluated changes to the probability density function (PDF) of the SPI and SPEI to determine changes to the distribution of the different drought events. This was further applied to assess the changes to the percentage of area under drought for the four NRM super-clusters assessed.

## 3 Results

### 3.1 Validation of Projected Droughts

We compared differences between CCAM derived metrics of droughts and those derived from observational products for the historical period (1981-2010) to quantify the biases of the historical simulations. The metrics derived from historical simulations for individual model runs tended to over-estimate SPI-based metrics and underestimate SPEI-based metrics, when compared against observational data (Fig. 2). The variability of biases across individual model runs was expected as heterogeneous runs from host GCM models were selected to estimate future model uncertainty. However, biases to the ensemble average were substantially reduced, denoting a good match to the observational data, particularly for SPEI.

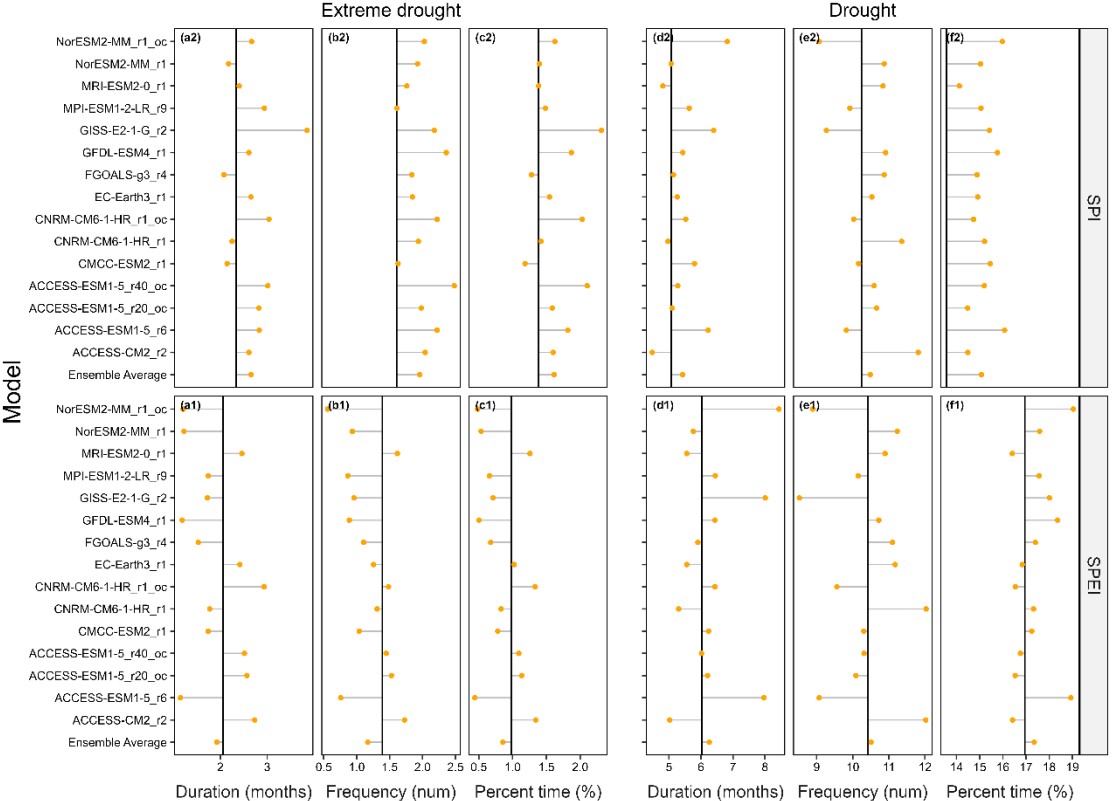

**Figure 2: Comparison of the differences in calculated metrics of drought and extreme droughts between climate model simulations and observations for the historical period over all of Australia (1981 to 2010). Drought metrics from observation data are presented as solid black lines, while points show metrics from climate model simulations.**

## 3.2 Climate Change Assessment

### 3.2.1 Changes to SPI and SPEI

The 20-year moving average SPI and SPEI time-series results under SSP370 are presented in Fig. 3. Decreases to SPEI were observed for all the models across all regions, indicating substantial agreement on future drying using SPEI. The largest decreases were observed by the end of the century. By contrast, the results for SPI were more heterogeneous, with many models predicting increases and decreases, as evident by the spread of models in the direction of trend (Fig. 3), though the ensemble averages tended towards a slight increase in wetness for the Rangelands and an increase in dryness for Southern Australia. These same patterns of change can be noted in the raw timeseries results of the ensemble averages presented in the supplementary materials for each emissions scenario (Fig. S3 to Fig. S5). Interannual variability from the different projections in each of the regions are presented in Fig. S25 to Fig. S48.

The time taken for the ensemble average to reach a 10% and 20% shift of the probability towards dryer conditions (according to the Z-score) are shown by vertical dashed lines. These thresholds were not reached for SPI using the ensemble average (though they are for some individual models) and hence no vertical dashed lines are shown. For SPEI a 10% shift towards drier conditions was reached by 2040 for the Rangelands and Southern Australia, and a 20% shift by 2060. These shifts of 10 and 20% were delayed in Northern Australia and Eastern Australia to approximately 2060 and 2090, respectively. Results for SSP126 and SSP245 are available in the supplementary materials (Fig. S6 and Fig. S7).

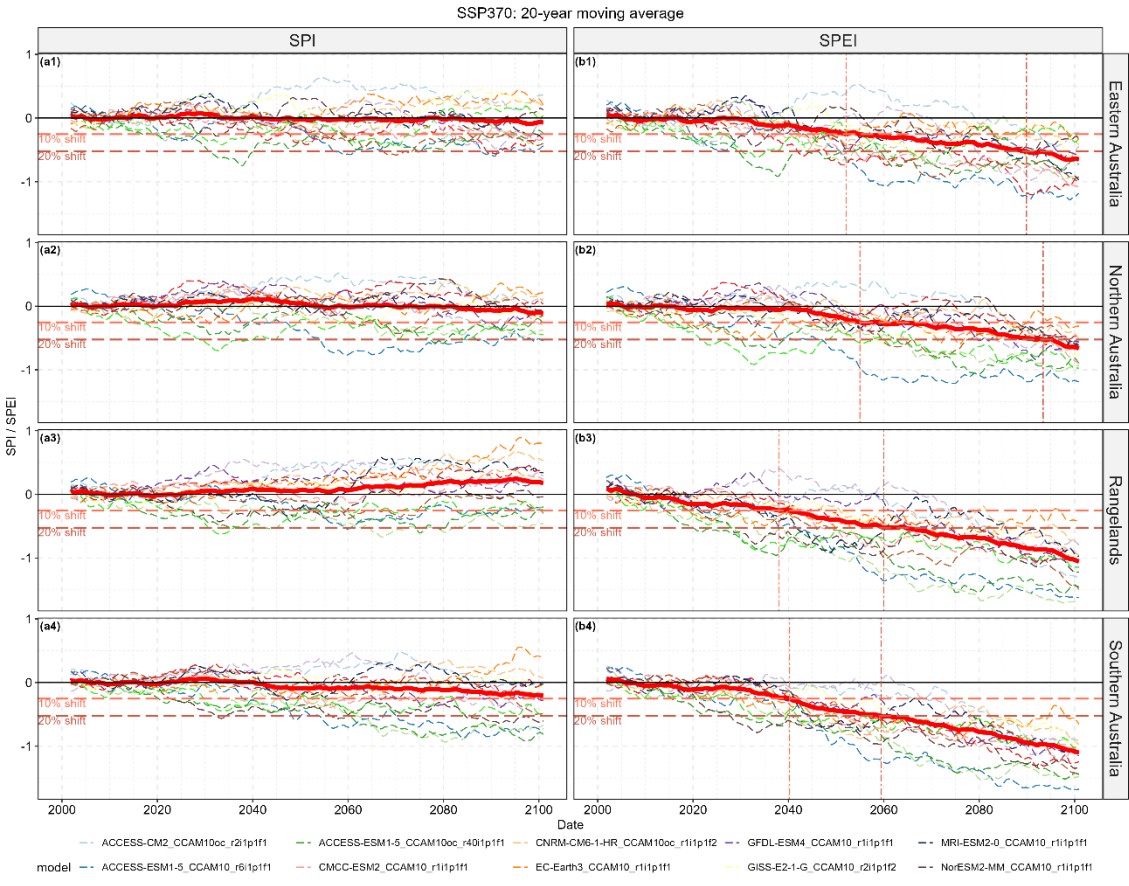

**Figure 3: Timeseries results for SPI and SPEI calculated as a 20-year moving average for each climate model considered with the ensemble average shown in red for each of the regions under the SSP370 scenario. Dotted lines show the time taken for the ensemble average value to shift by 10% and 20% (according to the Z-score).**

More wetting was evident under the high emissions scenario for the Rangelands compared to the low or moderate scenarios when considering only precipitation using SPI, but more drying when the additional impacts of increased PET were considered through SPEI (Fig. 4). For SPEI, all emissions scenarios consistently predict a 10% shift in the moving average value by approximately 2040, and a 20% shift by approximately 2060. Only at the end of the century, were there significant differences in SPEI between the different emissions scenarios, with the greatest decreases noted under SSP370. Similar patterns were also observed for the other NRM super-clusters assessed (Fig. S8 to Fig. S10).

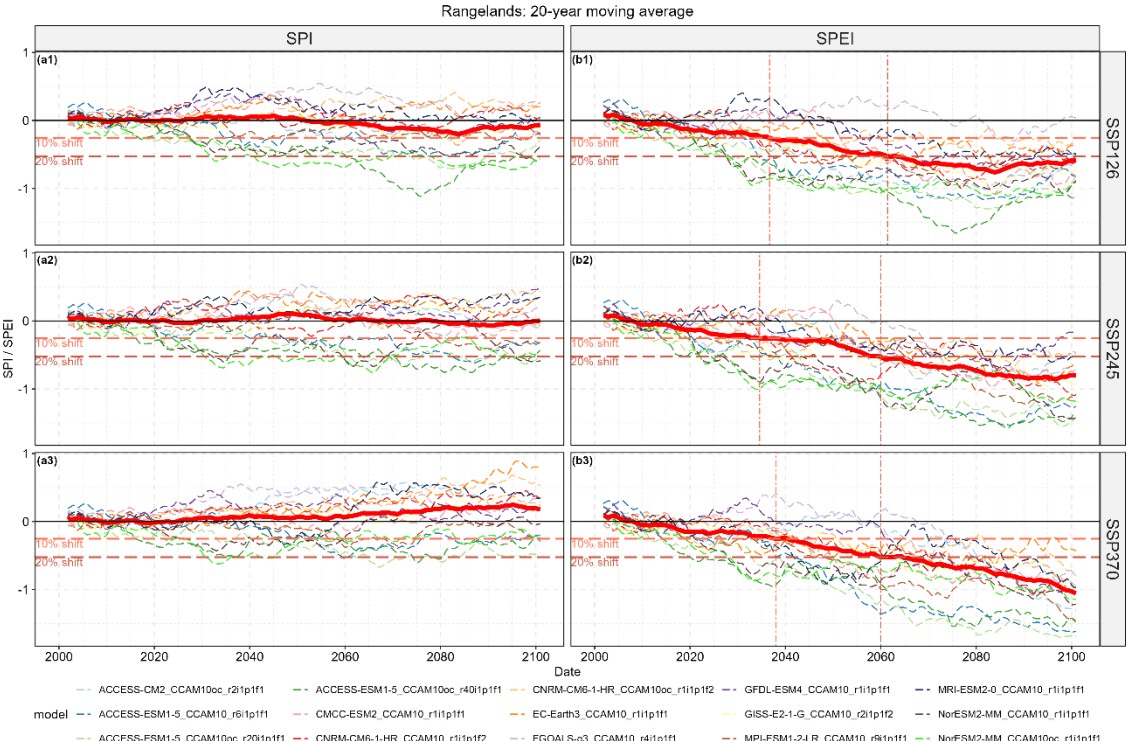

**Figure 4: Timeseries results for SPI and SPEI calculated as a 20-year moving average for each climate model considered with the ensemble average shown in red for each of the emissions scenarios for the Rangelands. Dotted lines show the time taken for the ensemble average value to shift by 10% and 20% (according to the Z-score).**

There was a notable shift towards more pronounced drought conditions in the 2050s and 2090s compared to the reference period (1995-2014) when assessing the probability density function (PDF) of both SPI (Fig. 5) and SPEI (Fig. 6). Relatively minor changes to the PDF were noted for SPI in Eastern Australia and Northern Australia, though there was a tendency towards lower SPI values (increased dryness) by the 2050s and 2090s compared to the reference period (1995-2014). Decreases were more pronounced for Southern Australia, while the changes to the Rangelands appeared minimal. In all regions, the largest changes were noted for the negative tails of the SPI distribution (< -1), indicating an increased likelihood of more pronounced periods of moderate to extreme droughts. Interestingly, in most regions, this appears to have come at the cost of the near normal and moderate wet categories (-1 to 1.5) but does not look to have changed the positive tail of the distribution (> 1.5) to the same extent. A quantification of the change to the probability of occurrence for the different categories of events under SSP370 confirms that the increase in extreme and severe droughts primarily led to decreased near normal and moderately wet conditions (Table 3). The probability of extreme wetness is shown to have also increased in all regions using SPI. This suggests an overall shift towards more periods of drought, while maintaining similar levels or increased periods of pronounced wetness (Fig. 5). There was an overall shift away from typical climate conditions towards more periods of both extreme drought and wetness (Table 3).

When the additional impacts of increased evaporation are considered using SPEI, there were notable shifts towards dryer conditions in all regions, especially by the end of the century (Fig. 6). This was particularly true for the Rangelands and Southern Australia (Table 3), which are subject to low rainfalls and therefore more strongly influenced by relative increases to PET. The shifts towards lower SPEI values and dryer conditions were seen across the full distribution of data, including the tails suggesting a future decrease to periods of wetness which was not reflected in the SPI results. Though only minor changes were projected for extreme wetness under SSP370 (Table 3). changes are shown to be considerably smaller under the moderate and low emissions scenarios (Table S2 and S3).

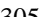

Figure 5: Probability density function plot of SPI values from the full ensemble of climate models for the reference period (1995-2014), 2050s (2041-2060), and 2090s (2081-2100). Results are shown for the three SSPs in the four NRM super-clusters considered. Dotted lines show mean values.

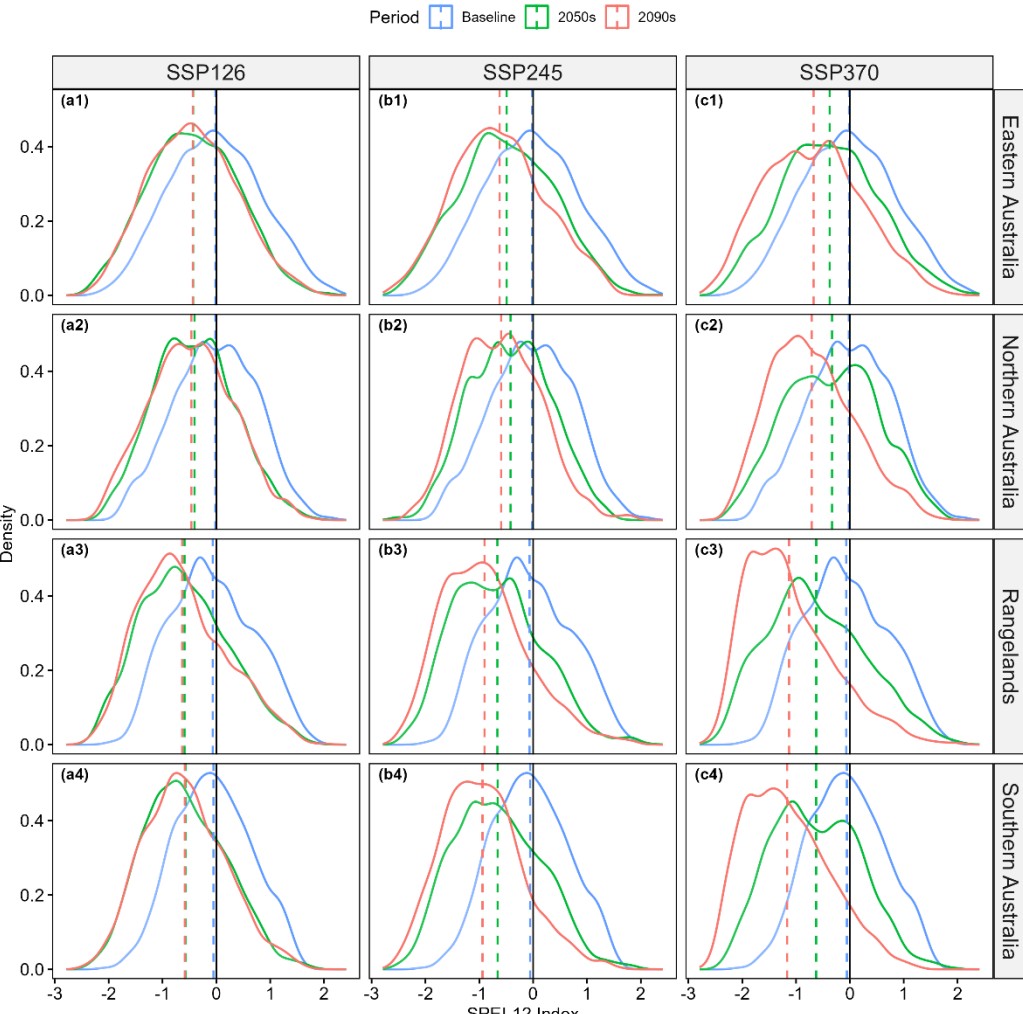

**Figure 6: Probability density function plot of SPEI values from the full ensemble of climate models for the reference period (1995-2014), 2050s (2041-2060), and 2090s (2081-2100). Results are shown for the three SSPs in the four NRM super-clusters considered. Dotted lines show mean values.**

**Table 3. Projected absolute percent change to the percent of time spent in different drought and wetness categories by the 2050s and 2090s compared to the reference period (1995-2014) using the ensemble average under SSP370. Red colours denote larger increases, while green colours denote decreases.**

| Index | Category | Eastern Australia 2050s | Eastern Australia 2090s | Northern Australia 2050s | Northern Australia 2090s | Rangelands 2050s | Rangelands 2090s | Southern Australia 2050s | Southern Australia 2090s |
|---|---|---|---|---|---|---|---|---|---|
| SPI | Extreme drought | 0.75 | 1.41 | 1.37 | 1.52 | 0.68 | -0.26 | 2.93 | 5.74 |
| | Severe drought | 0.48 | 1.49 | 1.22 | 1.35 | 0.46 | -0.78 | 2.07 | 2.98 |
| | Moderate drought | 0.08 | 1.22 | 0.59 | 1.71 | -0.15 | -1.82 | 1.42 | 1.84 |
| | Near normal | 0.01 | -3.84 | -4.36 | -3.38 | -3.76 | -2.39 | -7.33 | -11.37 |
| | Moderate wetness | -1.13 | -1.1 | -0.61 | -1.63 | -0.13 | 1 | -0.88 | -1.46 |
| | Severe wetness | -0.49 | -0.08 | 0.35 | -0.46 | 0.7 | 1.41 | -0.02 | 0.1 |
| | Extreme wetness | 0.3 | 0.9 | 1.43 | 0.89 | 2.2 | 2.84 | 1.81 | 2.17 |
| SPEI | Extreme drought | 3.7 | 9.88 | 4.29 | 8.23 | 8.71 | 20.99 | 8.88 | 24.78 |
| | Severe drought | 3.05 | 7.81 | 4.06 | 8.8 | 6.4 | 10.73 | 6.6 | 9.67 |
| | Moderate drought | 2.66 | 5.29 | 1.91 | 6.26 | 2.9 | 3.57 | 3.66 | 2.55 |
| | Near normal | -4.08 | -14.6 | -7.39 | -14.71 | -12.97 | -26.23 | -13.71 | -27.98 |
| | Moderate wetness | -3.13 | -5.13 | -2.43 | -5.55 | -3.88 | -6.68 | -4.02 | -6.52 |
| | Severe wetness | -2.23 | -3.27 | -1.1 | -2.96 | -1.99 | -3.49 | -2.29 | -3.75 |
| | Extreme wetness | -0.13 | -0.43 | 0.56 | -0.32 | 0.55 | -0.43 | 0.56 | -0.36 |

### 3.2.2 Changes to Drought Extent

A notable increase in the area affected by droughts was projected for all regions under SSP370 considering SPEI, with the largest increases noted by the end of the century and for Southern Australia and the Rangelands (Fig. 7). This same increase in drought extent, however, was not seen for SPI except in Southern Australia, where there was a trend towards more extreme droughts, though the magnitude of the change was significantly smaller than that seen for SPEI. Interestingly, the largest increases to drought extent occurred for extreme and severe events, while the extent of moderate droughts which are a more common occurrence under present conditions, did not increase significantly for either SPI or SPEI. These results suggest that the largest increases to droughts will occur for extreme events, rather than moderate events (Fig. 7 and Table 3). This is especially true when the impacts of increased PET are considered using SPEI. The results for SSP245 and SSP126 show more modest increases to drought extents for all the NRM super-clusters (Fig. S8 and Fig. S9), especially for the area in extreme drought, though the pattern of change remains the same.

PDFs of the area affected by extreme droughts are presented for SPI (Fig. 8) and SPEI (Fig. 9). For SPI, an increase to the area affected by extreme droughts can be seen in all regions and emissions scenarios, except for in the Rangelands under SSP370, where a minor decrease was projected by the end of the century (Fig. 8). These increases are typically in the order of 1 to 2%

of the area, representing a near doubling of the total area affected by extreme droughts. The increase was especially significant

in Southern Australia, where the average extent of extreme drought increases from 1.9% in the reference period to between 4.3 and 5.0% by the 2050s and 4.0 and 7.8% by the 2090s, depending on the emission scenario adopted. Under the high emissions scenario, this represented a fourfold increase to the area under extreme droughts. The magnitude of the changes were even more pronounced for SPEI, changing from 1.6% in the reference period to between 8.8 and 10.6% by the 2050s and 8.1 and 27.9% by the 2090s, depending on the emission scenario adopted (Fig. 9).

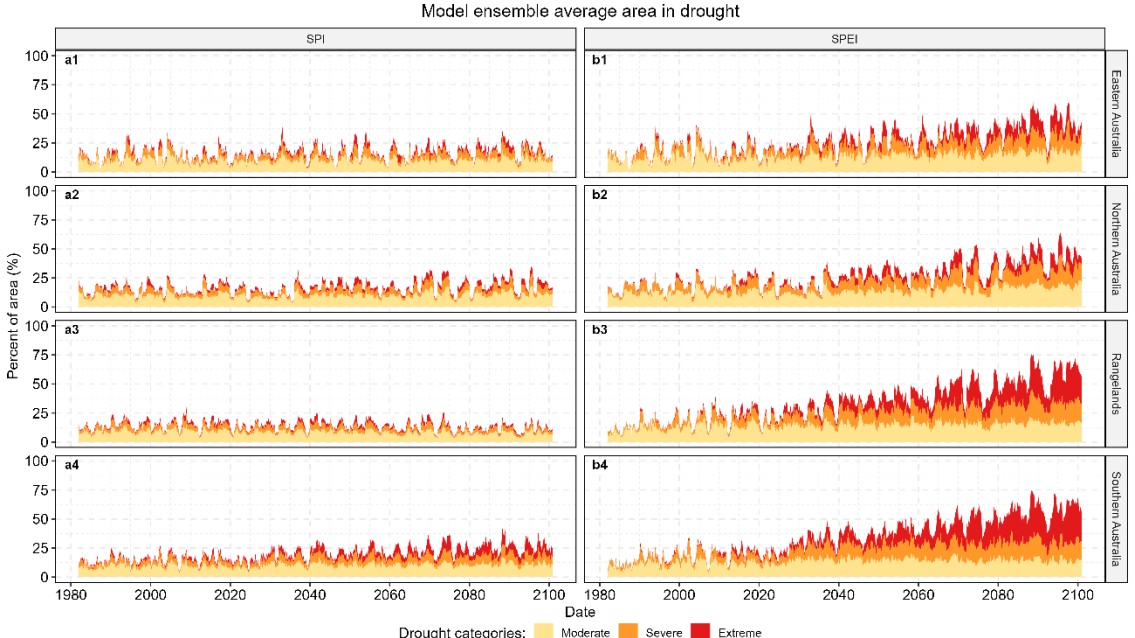

Figure 7: Timeseries of the ensemble average percent of area in drought in the four NRM super-clusters for SPI and SPEI under ssp370.

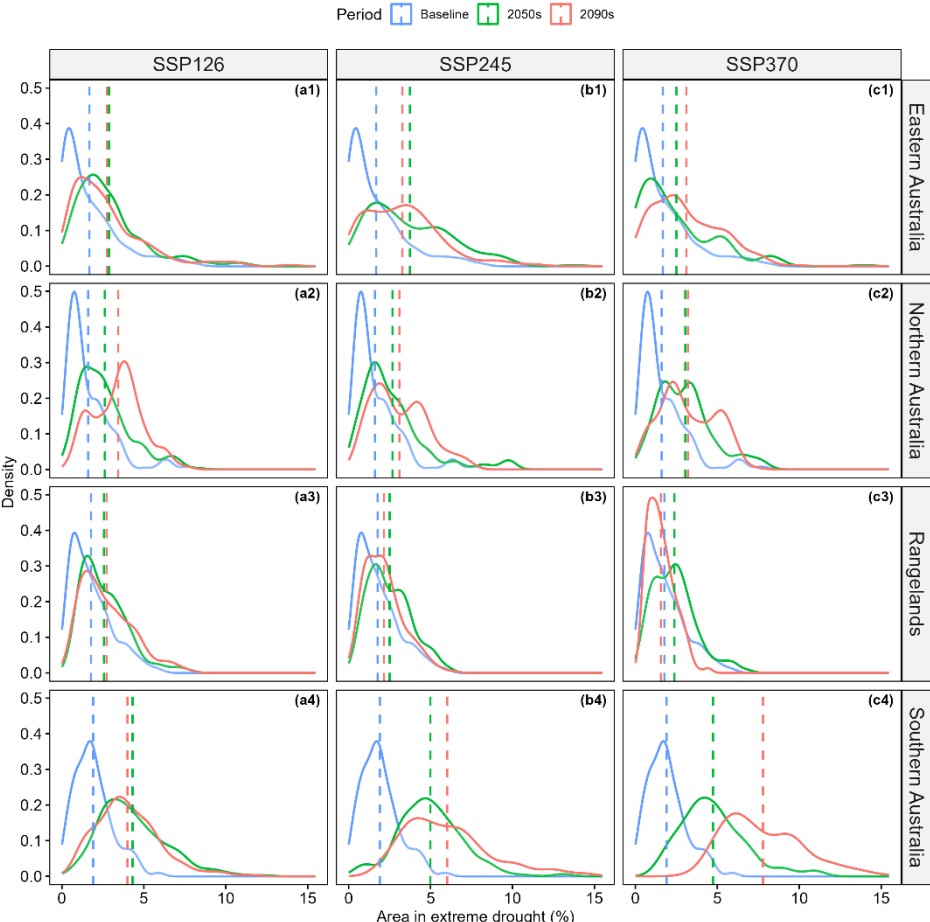

**Figure 8:** Probability density function plot the percent area under extreme drought using SPI-12 values from the ensemble average for the reference period (1995-2014), 2050s (2041-2060), and 2090s (2081-2100). Results are shown for the three SSPs in the four NRM super-clusters considered. Dotted lines show mean values.

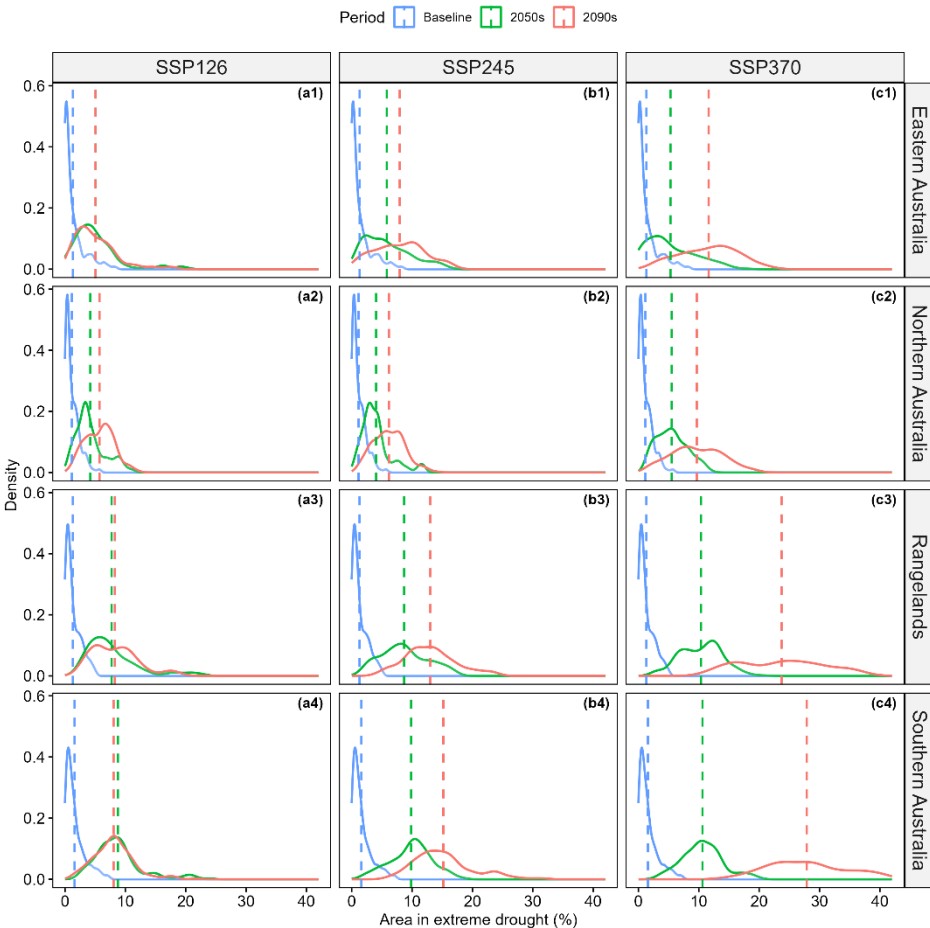

**Figure 9: Probability density function plot the percent area under extreme drought using SPEI-12 values from the ensemble average for the reference period (1995-2014), 2050s (2041-2060), and 2090s (2081-2100). Results are shown for the three SSPs in the four NRM super-clusters considered. Dotted lines show mean values.**

### 3.2.3 Changes to Drought Occurrence

For the percent time in drought, frequency, and duration of extreme droughts, there were few regions where the signal-to-noise ratio was greater than one for SPI (Fig. 10). Significant increases can be noted in south-west Western Australia, in southern Victoria, southern South Australia and in western Tasmania under the high emissions scenario (SSP370), which are seen to reflect the spatial changes of mean precipitation (Fig. S2). In southwest Western Australia, SPI related extreme droughts were

360 projected to occur both more frequently and last longer, leading to considerable increases in the percent time in drought. By contrast, the increases to the percent time in drought in southern Victoria, southern South Australia and in western Tasmania appears to be principally the result of increased drought frequency, with less clear changes noted for drought duration. In

addition to these regions, there were also significant increases to the percent time in moderate to extreme drought for the Gulf of Carpentaria and Northeastern Queensland for SSP370 by the 2090s (Fig. S15), which was not evident in the extreme

droughts. For the remainder of the country, the results of SPI tended to be more uncertain. Interestingly, there were no regions of Australia where there was a significant reduction to the time spent in extreme drought.

For SPEI, there was wide model agreement for more frequent and longer drought events for the majority of the continent, particularly under SSP370 and for the end of the century (Fig. 10). This was especially true for the percentage time in drought, which is the result of both increasing drought frequency and duration. For parts of Northern Australia and Eastern Australia,

there was generally less model agreement from the signal-to-noise ratio (as shown by the hatching) and the magnitude of the changes were typically smaller when compared to southern regions and the interior of the continent. There was a large range between the 10th and 90th percentile ensemble projections for both SPI and SPEI (Fig. S16 to Fig. S21), highlighting the uncertainty in these projections.

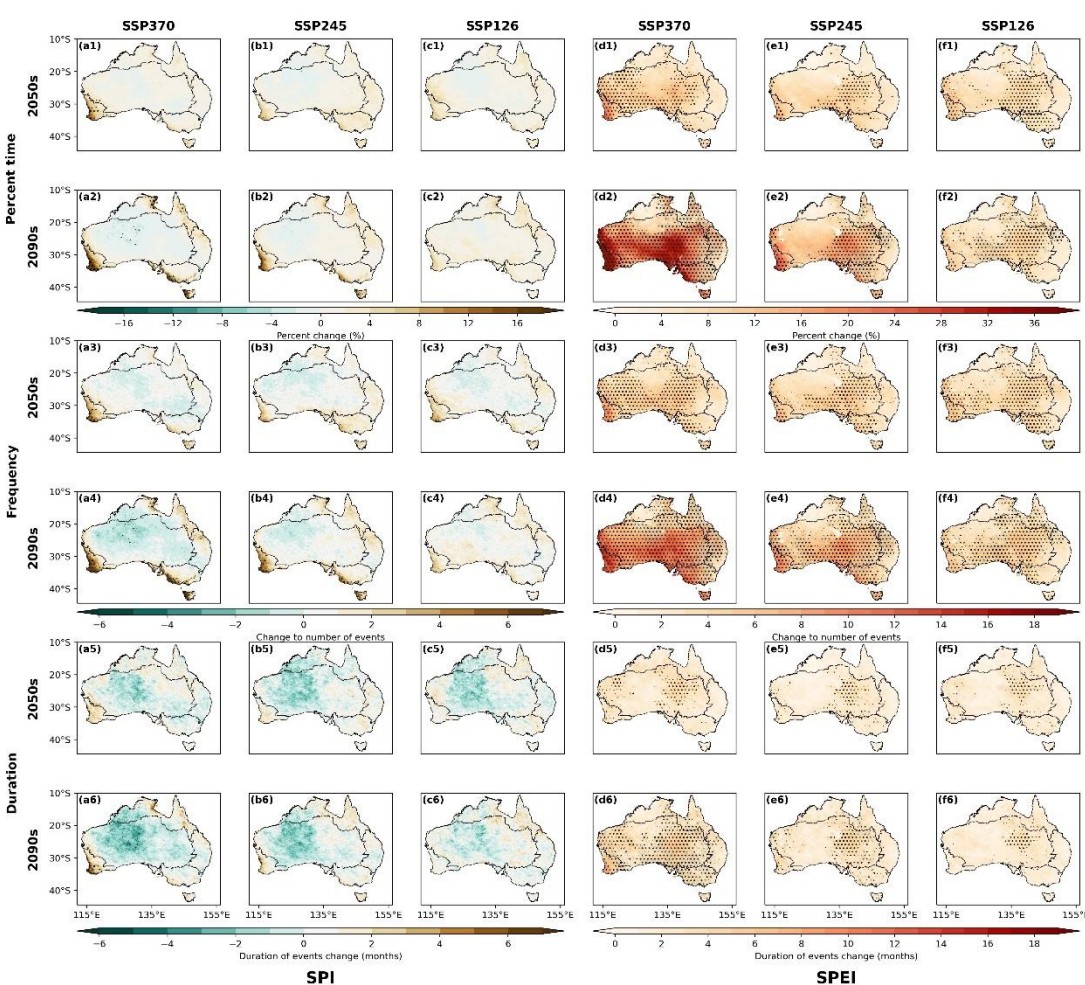

**Figure 10: Maps showing changes to the percent time (rows 1 to 2), frequency (rows 3 to 4), and duration (rows 5 to 6) of extreme droughts according to SPI (columns a, b, and c) and SPEI (column d, e, and f) for the 2050s and 2090s relative to the reference period. Hatching shows where the signal-to-noise ratio > 1.0.**

Considerable inter-model variability was evident in the projections as shown by boxplots from the model ensemble (Fig. 11), especially for SPEI. The variability was largest for the percent time in drought and frequency of droughts in the more arid regions of Southern Australia and Rangelands. While for drought duration, the model variability was greater in the more humid regions of Northern and Eastern Australia. The inter-model variability appears to approximately scale with the mean change in the projections, indicating greater uncertainty for larger changes. When using SPEI there was very wide agreement towards

more frequent and longer extreme droughts from the full ensemble of models in all regions. For SPI there was less certainty on the sign of change in most regions, except for Southern Australia where there was a clear tendency towards more frequent and longer extreme droughts. For Southern Australia, there was agreement between SPI and SPEI on the sign of the change but not the magnitude. For the other regions, the results were less certain, though generally most models appeared to point towards more frequent extreme droughts, with an overall increase to the time spent in extreme droughts for all regions and

emissions scenarios.

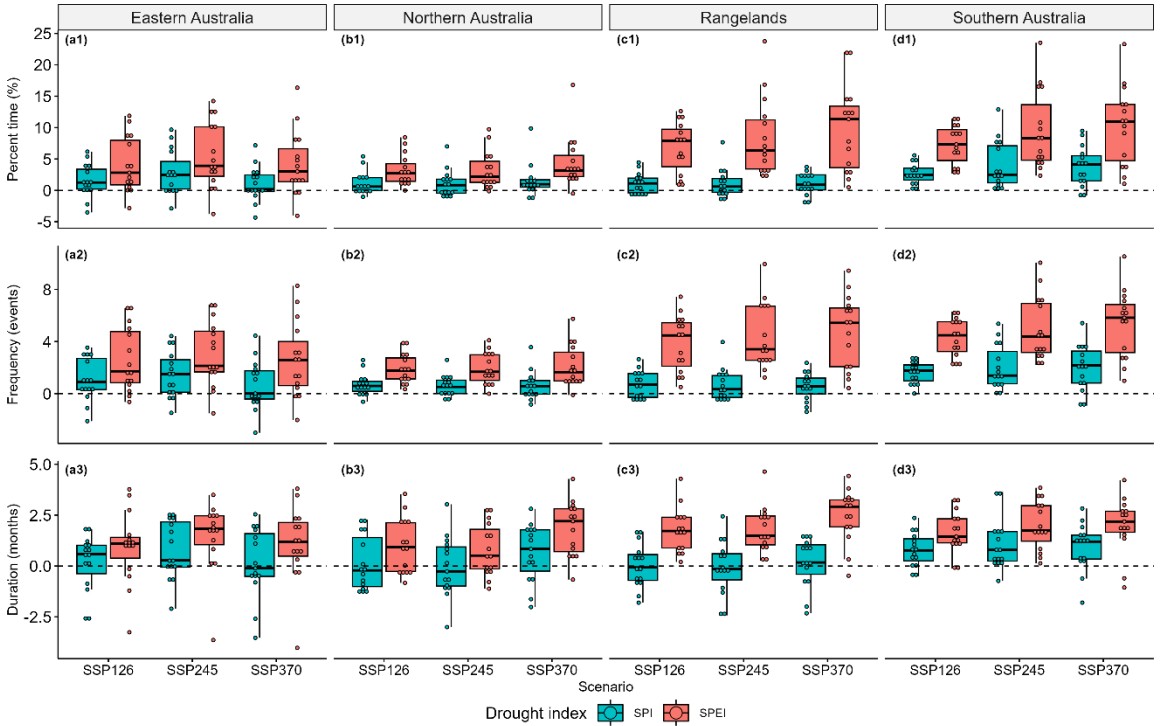

**Figure 11: Changes to the percent time, frequency, and duration of extreme droughts using SPI and SPEI in the 2050s compared to the reference period. The box and whisker plot shows the interquartile range (box), and the median (bar), while the whiskers extend**
**from the box to the furthest datapoint within 1.5x the interquartile range. Dots show projections for each of the climate models.**

## 4 Discussion

### 4.1 Future Drought

Our study shows there is likely to be an increase in the frequency of droughts, particularly extreme droughts, across Australia,
especially in Southern Australia and when assessing SPEI derived drought metrics. The results for SPI were more uncertain in
terms of the sign of change, reflecting uncertainty in rainfall projections (Fig. S2). Both drought indices projected an increase
to the percentage of time spent in drought as well as the spatial extent, frequency, and duration of droughts in southwest
Western Australia, southern South Australia, southern Victoria, and in western Tasmania, especially by the end of the century
and under high emissions (Fig. 10). While the sign of the change is clear in these regions, especially for SPEI, there is
considerable inter-model variability in the magnitude of the projected changes (Fig. 11), which may necessitate decision
makers to adopt an adaptive approach to planning for these future eventualities. These results are consistent with recent
observations which have pointed towards a trend of decreasing precipitation for these regions (Dey et al., 2019), and are also
consistent with recent global and regional assessments of future droughts (Cook et al., 2020; Herold et al., 2021; Kirono et al.,
2020; Spinoni et al., 2020; Ukkola et al., 2020; Wang et al., 2021; Zeng et al., 2022). Using earlier CMIP5 projections, Kirono
et al. (2020) showed a marked increase for future droughts in Southern Australia, which is in line with the findings from this
study. However, they also showed wide model agreement towards increased droughts in Eastern Australia using SPI, which
was not reflected in this study to the same degree. This may relate to the selection of the climate model ensemble adopted,
which has been shown to be one of the principal sources of uncertainty (Ukkola et al., 2018). Similarly, Trancoso et al. (2024)
have shown that the precipitation agreement of the host GCMs is particularly low for Australia for both CMIP5 and CMIP6
models, except for the southwest Western Australia region.

Our results show considerable increases to the area affected by future extreme droughts, especially in Southern Australia and
under the high emissions pathway. In the absence of strong mitigation of emissions (i.e. SSP370), an additional 5.9% increase
to the area affected by extreme drought was expected using SPI in Southern Australia by the end of the century, which
corresponds to a fourfold increase in the area affected compared to current conditions. Under a low emissions scenario
(SSP126), these increases are reduced to 2.1% or a near doubling compared to current conditions (Fig. 8). Differences between
emission scenarios were greater when evaluating the results of SPEI. Here, we found cutting emissions from high to low levels
by the end of the century would decrease the area affected by extreme droughts by a factor of 4 in Southern Australia, 3.2 in
Rangelands, 1.9 in Northern Australia, and 2.8 in Eastern Australia (Fig. 9), highlighting the importance of meeting emission
reduction targets. The increases to extreme droughts are larger than those projected for moderate droughts, particularly in
Southern Australia and the Rangelands (Table 3). Extreme droughts have a disproportionate impact on agriculture, society,
and the environment compared to more moderate droughts (Noel et al., 2020; Potop, 2011), and as such these changes would
likely necessitate robust adaptation measures. We provide supplementary datasets tailoring these projections to Australian

River Basins and Local Government Areas (Eccles, 2024). These datasets provide derived drought metrics at a much more granular scale, which may be useful for informing local and regional scale decisions on adaptation and drought preparedness.

Interestingly, the increase in extreme droughts did not lead to a decrease in extreme wetness, but rather mostly reduced time in near normal climate conditions (Table 3). Indeed, in some regions there was an increase to the time spent in extreme wet conditions in the future, indicating an overall shift towards more extreme climatic conditions. This was due to a shift in the mean and an overall flattening of the PDFs of SPI and SPEI as seen in Fig. 5 and Fig. 6, leading to more time in drought conditions. Similar PDFs changes have been noted in global assessments of soil moisture, runoff, and the Palmer drought index

under CMIP5 and CMIP6 (Zhao and Dai, 2015, 2022).

While there was wide model agreement on increased droughts for Southern Australia, our results point to less agreement among the ensemble of climate models and between the two drought indices for the other regions assessed. The differences between the two drought indices were particularly notable, with SPEI tending towards increased droughts for the majority of the continent, while results from the precipitation-based SPI were more uncertain (Fig. 10). The differences between SPI and

SPEI diverged further as the projections extended further into the future, with the largest differences noted by the end of the century and under the higher emissions scenario (Fig. 11), which corresponds to when atmospheric water demands from elevated PET were largest. Similar differences between these indices have been noted in studies using CMIP6 GCMs (Wang et al., 2021; Zeng et al., 2022). Atmospheric water demand was also found to be the principal factor contributing to increased future soil moisture drought over Australia (Zhao and Dai, 2022). Divergences between these indices have also been observed

in studies of the recent past, with the majority of the earth's landmass shown to have had a wetting trend using SPI between 1971 and 2022, and an opposing drying trend when evaluating SPEI (Nwayor and Robeson, 2023). For Australia, no trend was evident between 1980 and 2020 using SPI, while a significant drying was noted using SPEI (Vicente-Serrano et al., 2022).

### 4.2 Differences Between SPI and SPEI

Differences between SPI and SPEI were also more evident in arid and semi-arid regions such as the Rangelands, which receive

relatively low precipitation but have high potential for evaporative loss. In these regions, proportional increases to PET projected under climate change are substantially greater than the magnitude of possible changes to precipitation. As such, the relative impact of PET increases on the overall water budget (P – PET) is greater than in humid regions, where precipitation changes can be just as consequential. Precipitation variability has been shown to be the principal driver of SPEI in humid regions, while in arid regions PET is the principal driver (Vicente-Serrano et al., 2015). This is reflected in our projections of

future drought for SPEI, with smaller projected increases and less model agreement evident in the more humid Northern and Eastern Australia compared to the Rangelands and Southern Australia (Fig. 10 and Fig. 11). However, further PET increases which drive SPEI in water-limited regions (Rangelands and Southern Australia) are unlikely to have as much consequence as in humid regions where the potential upper limit of actual evaporation has not already been met.

In this study, PET was derived using the Penman-Monteith method (Allen et al., 1998). This approach is more data intensive

than simplified techniques that rely on temperature inputs only, but is considered more robust and has been recommended

when data is available (Hosseinzadehtalaei et al., 2017; Sheffield et al., 2012). Purely temperature-based models such as Thornthwaite (Thornthwaite, 1948) and Hargreaves (Hargreaves and Samani, 1985) equations have been shown to overestimate future PET. A limitation of this approach is that the approach for deriving PET does not resolve interactions between elevated $CO_2$ and vegetation (Trancoso et al., 2017). Specifically, studies have shown that elevated $CO_2$ results in reduced stomatal conductance and elevated water use efficiency of vegetation (Leakey et al., 2009), leading to reduced transpiration (Novick et al., 2016). However, increased fertilisation from elevated $CO_2$ would likely lead to increased leaf size (Pritchard et al., 1999) and increase transpiration.

While there is some disagreement on the magnitude of future PET increases, there is confidence in the sign of change, unlike for precipitation for which there is much uncertainty around the sign of future changes (Trancoso et al., 2024). Under climate change, increasing temperatures will lead to increased evaporative demand, impacting on the overall water budget. Studies which adopt SPI only to assess future changes to droughts miss this important component and may therefore underestimate future drought changes. On the other hand, there is potential that the SPEI could overestimates future drought magnitudes, especially in water-limited regions and would rather represent a conservative upper limit of potential future drought risk. Changes to other drought types may therefore end up lying somewhere between these two indices, depending on the drought type and the region assessed (Reyniers et al., 2023; Tomas-Burguera et al., 2020).

The simulated changes to drought are likewise influenced by the projected land cover changes incorporated into CCAM as part of the emissions scenarios (Eyring et al., 2016). These landcover changes are not dynamic or responsive to changes in the climate but rather follow prescribed changes from one land cover type to another. The changes in land cover can influence temperatures and windspeed (due to changing surface roughness) in the projections, therefore influencing PET and SPEI in some regions.

## 4.3 Implications

While this study focused only on meteorological droughts, these changes will have inevitable consequences for other drought types (e.g. agricultural and hydrological), though it should be noted that the propagation from meteorological droughts to other drought types is typically non-linear (Mukherjee et al., 2018). It should be noted that increases to SPEI may not necessarily translate into on the ground changes, especially in water-limited environments where PET is already far greater than precipitation. In these regions, which includes most of Australia the timing and magnitude of precipitation may be a more important consideration, and as such care must be taken when interpreting the SPEI-based drought projections. Significant decreasing trends for streamflow have been observed for most of Australia in the recent past, with only catchments in the northern tropics showing an increasing trend (Amirthanathan et al., 2023). This has led to increased hydrological droughts over much of southern Australia, which cannot be explained by changes to rainfall alone (Wasko et al., 2021). In Southeast Australia, the Millenium drought (2001-2009) was a major contributor to decreased streamflow (Fiddes and Timbal, 2016). However, despite the meteorological drought breaking in 2010, a hydrological drought has persisted in many catchments, with runoff volumes significantly lower than pre-drought conditions despite a return in precipitation (Fowler et al., 2022; Peterson

et al., 2021). This suggests that hydrological droughts can persist indefinitely following prolonged meteorological droughts
(Peterson et al., 2021). Future increases to the time spent, extent, and duration of meteorological droughts as suggested by this study may therefore have significant ramifications for hydrological droughts in Australia, by effectively altering the long-term rainfall-runoff response. In southwest Western Australia, observed streamflow declines have been attributed to a combination of decreased rainfall and increased vegetation (Liu et al., 2019). $CO_2$ fertilisation may therefore work in tandem with meteorological droughts to further exacerbate future hydrological droughts (Mankin et al., 2019; Trancoso et al., 2017) in spite
of $CO_2$ induced changes to stomatal conductance reducing plant transpiration changes.

Both positive and negative changes in landcover can influence meteorological droughts through changes in precipitation, temperature, and windspeed (due to changing surface roughness). For instance, in southwest Western Australia largescale anthropogenic landcover changes were shown to partially drive long-term declines in precipitation along coastal regions and increases in inland regions (Pitman et al., 2004; Timbal and Arblaster, 2006). The projections included in this study incorporate
time varying landcover changes which are prescribed according to the emissions scenario (Eyring et al., 2016), though these are relatively minor for Australia. These changes are, however, not dynamic or responsive to changes in the climate and as such could respond differently in the future, potentially impacting on the magnitude of the drought changes presented. It is important to note that such changes to landcover and other associated environmental factors (e.g. groundwater and soil moisture) would have much more profound impacts for other drought types (e.g. agriculture and hydrological) compared to
meteorological droughts as these are directly influenced by land surface characteristics.

Elevated PET during periods of precipitation deficit will likely increase the severity of plant stress due to differences between the atmospheric water demand and the water available for transpiration (Anderegg et al., 2015). This can lead to plant dieback and mortality, which may also be worsened from elevated heat stress due to a warming climate. potentially influencing the propagation and response of future droughts. Higher atmospheric water demand can also work to dry out vegetation and elevate
fire risk (Clarke et al., 2022). The recent tinderbox drought in southeast Australia is an example of a drought characterised by below average rainfall, high atmospheric water demand, and reduced water availability (Devanand et al., 2024). The high atmospheric water demand and limited water availability led to elevated temperatures, amplified heatwaves, and likely contributed to the Black Summer bushfires (Devanand et al., 2024). An amplification of future meteorological droughts characterised by elevated PET and higher temperatures may therefore lead to an increase in such events, which will have
obvious ramifications for bushfire risk and heatwaves. Further research is, however, required to quantify the magnitude of these future changes as a result of the projected meteorological drought changes.

**5 Conclusion**

We evaluated the impacts of climate change on meteorological droughts using two commonly adopted indices (SPI and SPEI). For this purpose, high-resolution CMIP6 climate models under three SSP scenarios were applied. The results show consistent
increases in future frequency, duration, percent time, and spatial extent of SPI droughts for south-west Western Australia,

southern Victoria, southern South Australia, and in western Tasmania, while a majority of Australia was projected to see increases according to SPEI. The increases were largest by the end of the century and under the high emission (SSP370) scenario, especially for SPEI, as this is when increases to temperature and evaporative demand were greatest. These increases appear to have largely come at the expense of 'normal' climatic conditions, with little changes or small increases to time spent under extreme wet conditions, pointing towards an overall shift towards more extreme climatic conditions across Australia. There was greater certainty on the sign of change for droughts when assessing SPEI compared to SPI for all regions due to strong certainty of increasing PET, though there was still considerable uncertainty on the magnitude of the changes. Under a scenario of high emissions, a 4-fold increase in the area affected by extreme drought was expected for Southern Australia by the end of the century, considering just changes to rainfall (SPI). When the additional impacts of evaporative losses from PET were considered (SPEI), there was a 17-fold increase in the area impacted compared to current conditions. Under a low emissions scenario, these changes decreased to 2-fold for SPI and 5-fold for SPEI, highlighting the importance of mitigating emissions. The relative changes were less substantial for the other NRM region clusters assessed, except for the Rangelands for which significant increases were shown when evaluating SPEI by the end of the century but not for SPI. Overall, our findings show strong increases in meteorological droughts for the majority of Australia, particularly in the southern region, by the end of the century, and under high emissions scenarios. These results have multi-sectoral implications with strong impact on water supply and agriculture and we encourage stakeholders to explore the supplementary datasets with tailored drought calculations for Australian Local Government Areas and River Basins to support decision-making.

## Data availability

Datasets of regionalised drought changes are freely available from (Eccles, 2024, https://doi.org/10.6084/m9.figshare.26343823)

The downscaled CCAM data used in this study is being published via the CORDEX Australasia domain archive. The 20km data for the Australasian CORDEX domain is available from NCI (National Computational Infrastructure): https://dx.doi.org/10.25914/8fve-1910. Selected 10 km resolution data is available from NCI for registered users.

## Author Contribution

Rohan Eccles: Writing – Original draft preparation, Conceptualization, Methodology, Formal analysis, Ralph Trancoso: Conceptualization, Methodology, Writing - Review & Editing, Jozef Syktus: Conceptualization, Data Curation, Methodology Sarah Chapman: Writing - Review & Editing, Data Curation, Visualization, Nathan Toombs: Writing - Review & Editing, Data Curation, Hong Zhang: Writing - Review & Editing, Data Curation, Shaoxiu Ma: Writing - Review & Editing, Ryan McGloin: Writing - Review & Editing

## Competing Interests

The authors declare that they have no conflict of interest.

## Acknowledgements

We acknowledge support by Lindsay Brebber from Information and Digital Science Delivery of the Department of Environment and Science for support with high performance computing and data storage.

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
