# Peer review of "High-resolution downscaled CMIP6 drought projections for Australia"

_EGUsphere, 2024_

## Referee Comment (RC2)

The manuscript 'Meteorological Drought Projections for Australia from Downscaled high-resolution CMIP6 climate simulations' presents the future drought features (SPI and SPEI) based on the downscaled precipitation and potential evapotranspiration data. The work is well-presented. However, there are some issues that need to be clarified further before the publication.

1. This study utilizes various drought characteristics, including duration, frequency, percent time (Figure 2), and shifts in the moving average, to predict future droughts. However, since the downscaling is applied only spatially, all temporal analyses could be conducted using GCM data. Yet, only Figure 10 presents a spatial map. What is the rationale for using downscaled data in this context?

2. Why did the author choose to use downscaled data from the Conformal Cubic Atmospheric Model (CCAM)? What advantages does CCAM offer compared to other downscaled datasets? Additionally, how can you demonstrate that drought characteristics derived from the downscaled data are more reliable or accurate than those based on raw GCM data?

3. Is there any result about the comparison between the downscaled data and original data (such as precipitation and potential evapotranspiration) to evaluate the downscaling methods' performance?

4. The study area was divided into four distinct regions—Eastern Australia, Northern Australia, the Rangelands, and Southern Australia—based on climatic and biophysical characteristics. However, the specific climatic and biophysical parameters used for this classification were not explicitly defined. Including detailed information on climate patterns (e.g., precipitation regimes, seasonal variations), dominant vegetation types, and temperature ranges could enhance the clarity of the classification framework. Such specifications would facilitate a more comprehensive interpretation of the analytical results by providing critical contextual information about regional environmental variations.

5. The discussion's comparative analysis of the SPI and SPEI offers valuable methodological insights. However, stronger integration with region-specific climatic and biophysical drivers would benefit the interpretation. Additionally, the spatial specificity of distinctions between SPI and SPEI across sub-regions remains insufficiently delineated, limiting the granularity of conclusions.

   Can more spatiotemporal visualizations (e.g., seasonal or interannual variability in drought indices in different regions) be incorporated to elucidate sub-regional heterogeneity clearly?

   The discussion should also explicitly articulate linkages between index disparities and potential localized environmental drivers, such as land cover status.

---

## Community Comment (CC1)

**HESS** is an open-access journal. For your review, I have included an excerpt from the journal office's official website(https://www.hydrology-and-earth-system-sciences.net/)  below.

"Hydrology and Earth System Sciences (HESS) is a not-for-profit international two-stage **open-access journal** for the publication of original research in hydrology. HESS encourages and supports **fundamental and applied research** that advances the understanding of hydrological systems, their role in **providing water for ecosystems and society, and the role of the water cycle in the functioning of the Earth system**. A multi-disciplinary approach is encouraged that broadens the hydrological perspective and the **advancement of hydrological science through integration with other cognate sciences and cross-fertilization across disciplinary boundaries.**"

Why would the journal office entertain anonymous referees?  If the journal is an open-access journal, how could the anonymous referees' reviews fit into the journal's statement of openness?

Do the authors have rights to request the journal office to disclose the names of the referees who are scared to disclose their names?

Do the readers of the journal have rights to request the journal office to disclose the names of the referees who are scared to disclose their names?

Could it be possible for the journal office to have clerical staff to post reviews as invited referees? Is this what we expect from EU and its official statements?

What is the relationship between the handling editor and the invited referees? Are the referees paying the handling editors to get them invited?

Why would an invited referee be scared to disclose his/her name in an open-access journal? Is it because he/she is unqualified and lacks potential and expertise in his/her field of specialization to demonstrate in an open-access journal?

I am requesting the handling editor to disclose the reason for promoting anonymity in an open-access journal. I am also requesting the journal office to disclose the rationale behind appointing the handling editors and referees. I request the journal office to disclose academic transcripts of the handling editors and referees.  What did they do in their PhD work? Who were their supervisors? What were their GER scores in their postgraduate studies? How many billions have been invested in their PhD philosophies? These details are pertinent to showcase that the EU and the journal office don't favor appointing handling editors and referees in an open-access journal.

---

## Author Comment (AC1)

**Response to Reviewer 1 comments**

*General Comment:*

The paper describes a set of drought projections for Australia developed by dynamically downscaling CMIP6 global climate models. A possible range of future drought conditions is considered that span multiple emissions pathways and model configurations. Future drought conditions are described through event frequency, duration, spatial extent and time spent in drought, in terms of changes in two commonly used drought metrics. The paper is well-structured, the information is clearly presented and the key results appropriately discussed. I recommend publication after minor revisions, which I describe below.

*Response:*

We thank the reviewer for the time spent on our manuscript and for the positive and constructive comments provided. Our comments below indicate where we plan to make changes in the manuscript to satisfy these concerns.

*Comment:*

One key issue is the use of SPEI. This metric, being the difference between precipitation and potential evapotranspiration, is intended to reflect the atmospheric water balance and thereby give a complementary view to SPI-based drought. However, the use of potential evapotranspiration in the calculation of SPEI makes SPEI unrealistic in many water-limited parts of Australia, where actual evapotranspiration does not approach the potential upper limit. So, any projected worsening of PET-related conditions is merely an indication of an increase in atmospheric demand for moisture, rather than a conclusive reduction in water stores. I suggest this issue is more adequately discussed in the paper, including the implications in the interpretation of SPEI-based projections of drought.

*Response:*

We agree with the reviewer that this is a key limitation of the SPEI metric. We will strengthen our discussion of the interpretation of the SPEI-based drought projections. Specifically, as suggested by the reviewer we will expand the discussion of the implications (Section 4.3) to make it clear that increases to SPEI-based drought projections do not necessarily translate to an increase in agricultural and hydrological droughts in water-limited regions. We will also bring this point into the discussion of the differences between SPI and SPEI (Section 4.2).

*Comment:*

The second key issue is the lack of attention given to the uncertainty of the projections. While using a multi-model ensemble and multiple emissions pathways goes some way to addressing uncertainty, the drought projections should be presented along with quantified uncertainty estimates. Moreover, the issue of uncertainty propagation from GCM through to downscaling technique to RCM was not addressed.

*Response:*

We thank the reviewer for their suggestion. We have attempted to show the uncertainty from the multi-model ensemble and multiple emissions pathways using timeseries plots of all ensemble members (Figure 3 and 4), probability density function plots (Figure 5 and 6 and Figure 8 and 9), and boxplots (Figure 11). Our spatial maps, however, only show the multimodel average. To address this, we will present spatial maps of the 10th and 90th percentile of changes along with the multi-model average. Due to the number of maps involved these figures will be added to the revised supplementary materials.

We have previously evaluated how projections of mean climate and extremes compare from these downscaled projections to the host models (Chapman et al., 2024). This analysis showed there to be very good agreement between the host GCMs and CCAM for temperature and good agreement for precipitation. For precipitation the spatial patterns were generally preserved, however, there were some changes in magnitude in some seasons. It was not possible to compare the PET used in this study to the GCMs, as it was derived offline using several climate CCAM output variables as input data. We will update the Methodology (Section 2.2) in our revised manuscript with this added information to inform the reader.

*Comment:*

The final key issue is that one of the most crucial findings of the study needs to be made more prominent. The results show that more time is projected to be spent under extreme conditions, both wet and dry, and less time under 'normal' conditions, for some parts of Australia (Table 3). This result should be made more prominent, for example by featuring in the abstract. This result is important because it suggests that the combination of projected changes in the climate system is shifting the dial towards more extreme climatic conditions and motivates future research in understanding the physical processes responsible for the shift.

*Response:*

We thank the reviewer for pointing this out and agree that this could be better highlighted as one of the key findings. Specifically, we will add a sentence to the abstract and the conclusion highlighting how the increased time spent under drought appears to have largely come at the expense of 'normal' conditions, while there seems to be little change or increases to the time spent in extreme wet conditions, indicating an overall shift towards more extreme climatic conditions. Additionally, we will expand on our discussion of this finding within Section 4.1.

*Minor Comment:*

L84: this is a bit of a throw away line. I suggest turning this around by stating that since RCMs have been shown to estimate regional rainfall features with higher precision than GCMs, RCMs are more appropriate to study drought on the regional scale.

*Response:*

We agree with the reviewer and will change this line to the following in the revised manuscript: "However, while research to date has largely focussed on applying coarse GCM outputs to assess future droughts, RCMs have been shown to have more skill in representing key rainfall features and may therefore be better suited to study droughts at regional scales."

*Minor Comment:*

Inter-model variability (Figure 11) is shown to be higher for SPEI and some drought characteristics. Can an explanation be offered for why this is? What are the implications of this variability on the interpretation of future drought changes?

*Response:*

We thank the reviewer for this question. We believe the variability of the projected changes is

related to the mean projected change (i.e. the range of changes approximately scales with the mean change). For instance, we believe there is greater variability of projected changes for SPEI compared to SPI, as the mean changes to SPEI are larger. Additionally, the variability for SPEI is greater in water-limited regions (Rangelands and Southern Australia) compared to the more humid regions (Northern and Eastern Australia) as mean changes are also larger in these regions. To illustrate this point, we present the relationship between the mean change and the standard deviation of the change for all extreme drought metrics in all regions below, which appears to support this hypothesis. We will include some added discussion around these results in Section 4.1 of the revised manuscript.

[Figure]

References:
Chapman, S., Syktus, J., Trancoso, R., Toombs, N., & Eccles, R. (2024). Projected changes in mean climate and extremes from downscaled high-resolution CMIP6 simulations in Australia. *Weather and Climate Extremes*, *46*, 100733. https://doi.org/10.1016/j.wace.2024.100733

---

## Author Comment (AC2)

**Response to Reviewer 2 comments**

*General Comment:*

The manuscript 'Meteorological Drought Projections for Australia from Downscaled high-resolution CMIP6 climate simulations' presents the future drought features (SPI and SPEI) based on the downscaled precipitation and potential evapotranspiration data. The work is well-presented. However, there are some issues that need to be clarified further before the publication.

*Response:*

We thank the reviewer for their time and constructive comments on our manuscript. Our comments below show where we plan to make changes to the manuscript to address these concerns.

*Comment:*

1. This study utilizes various drought characteristics, including duration, frequency, percent time (Figure 2), and shifts in the moving average, to predict future droughts. However, since the downscaling is applied only spatially, all temporal analyses could be conducted using GCM data. Yet, only Figure 10 presents a spatial map. What is the rationale for using downscaled data in this context?

*Response:*

We thank the reviewer for their comment. It is important to note that downscaling does improve the temporal and the spatial resolution of the projections (for instance CCAM has sub-daily data available). However, as this analysis was conducted using accumulated monthly precipitation and PET, outputs from GCMs could also be applied as suggested by the reviewer, though at a much coarser spatial resolution. We have found in previous work that the downscaling does improve the representation of precipitation and temperature, even when assessed at coarse spatial scales (Chapman et al., 2023). The benefits are greater for coastal regions or where the terrain is complex. These improvements provide benefits to projections of future drought events even when assessed across the Natural Resource Management (NRM) regions, which are relatively coarse. We will make improvements to the introduction to make these benefits clearer in the revised manuscript. It is also important to note that as part of this paper we provide regionalised drought characteristics for Australian Local Government Areas and River Basin (https://doi.org/10.6084/m9.figshare.26343823), for which the finer granularity of the downscaled projections is very beneficial.

We made a conscious choice in the manuscript to combine the spatial maps into subplots where possible to allow for the changes to different drought characteristics to be interpreted spatially together between SPI and SPEI as this allows for an easier comparison of the differences. For instance, Figure 10 highlighted by the reviewer contains subplots of 36 maps for extreme droughts. Additional maps of changes to moderate droughts can be seen in the supplementary materials (Figure S14). Additionally, we will be including additional spatial maps of the 10[th] and 90[th] percentile of changes to better show the uncertainty of these changes.

 we will include more spatial visualisations of the 10[th] and 90[th] percentile of changes to the

*Comment:*

2. Why did the author choose to use downscaled data from the Conformal Cubic Atmospheric Model (CCAM)? What advantages does CCAM offer compared to other downscaled datasets? Additionally, how can you demonstrate that drought characteristics derived from the downscaled data are more reliable or accurate than those based on raw GCM data?

*Response:*

The reviewer is correct that there are other downscaled datasets available as part of the CORDEX CMIP6 experiment. However, at the time that this work was undertaken, only the CCAM dataset was available for analysis. It should also be noted that we adopted the reference crop evapotranspiration (PET) for calculating the SPEI, which was derived offline from CCAM. This requires some considerable effort as several variables are required at a daily timestep (daily data solar radiation, vapour pressure, maximum and minimum temperature, mean sea level pressure, and wind speed). As such, offline PET is not available for either the GCMs or other downscaled datasets, making a one-to-one comparison difficult. Lastly, CCAM is advantageous over other datasets, as it is the largest ensemble available (15 models) and run at the highest resolution (10 km).

The CCAM dataset has been previously evaluated against the host GCMs as part of an assessment of added value, which showed downscaling improved simulations of precipitation and temperature with added value of up to 150% across Queensland's regions (Chapman et al., 2023), especially for extremes and over regions with complex terrain. We will better highlight these advantages in the introduction and methodology of our revised manuscript. Our other paper (Chapman et al., 2024) also shows how high resolution projections add details to regional climate hazard analysis. We will better highlight these advantages in the introduction and methodology of our revised manuscript.

*Comment:*

3. Is there any result about the comparison between the downscaled data and original data (such as precipitation and potential evapotranspiration) to evaluate the downscaling methods' performance?

*Response:*

We thank the reviewer for their comment. As we note above, the downscaled precipitation data has previously been evaluated against observations and compared to the host models in an assessment of added value (Chapman et al., 2023). This analysis found that downscaling improved performance over host GCMs for seasonal temperature and precipitation (10% and 43% respectively), and for annual cycles of temperature and precipitation (6% and 13% respectively). Downscaling also improved the fraction of dry days, reducing the bias for too many low-rain days. As PET was derived offline from the model (as noted above), we could not compare the performance from CCAM and the host GCMs. We will better highlight these advantages in our revised manuscript.

*Comment:*

4. The study area was divided into four distinct regions—Eastern Australia, Northern Australia, the Rangelands, and Southern Australia—based on climatic and biophysical characteristics. However, the specific climatic and biophysical parameters used for this classification were not

explicitly defined. Including detailed information on climate patterns (e.g., precipitation regimes, seasonal variations), dominant vegetation types, and temperature ranges could enhance the clarity of the classification framework. Such specifications would facilitate a more comprehensive interpretation of the analytical results by providing critical contextual information about regional environmental variations.

*Response:*

We thank the reviewer for pointing out the lack of information regarding how the NRM regions were defined. It is important to note that we did not classify these regions ourselves. Rather, we adopt pre-defined regions which were developed by CSIRO and BOM to specifically assess climate change in Australia (CSIRO and Bureau of Meteorology, 2015). These regions are recommended for use in climate change studies of Australia and have been widely applied for this purpose (Chapman et al., 2024; Grose et al., 2020; Wasko et al., 2023), including for droughts (Kirono et al., 2020). We will update our manuscript to include the original reference in the methodology which details how they were defined (CSIRO and Bureau of Meteorology, 2015). As these regions are relatively large, there are a number of vegetation types and climate zones included within each one. We have collated some of the relevant information from the original report into a table as suggested by the reviewer (see below), which we will include in the updated supplementary materials.

Additionally, we will revise Figure 1 in the revised manuscript to include the major climate regions as a background so that these can be easily compared against the delineated NRM regions.

| NRM super-cluster | Area (1000 km²) | Climate Zone | Ecoregions |
| --- | --- | --- | --- |
| Eastern Australia | 767 | Subtropical (north) Temperate (south) Grassland (west) | Temperate broadleaf and mixed forests Temperate grasslands, savannas and shrublands Tropical and subtropical grasslands, savannahs and shrublands |
| Northern Australia | 2084 | Equatorial (north east) Tropical (north) Subtropical (far east) Grassland (south) | Tropical and subtropical grasslands, savannahs and shrublands Tropical and subtropical moist broadleaf forests |
| Rangelands | 4888 | Grassland (scattered) Desert (majority) | Deserts and xeric shrublands (majority) Mediterranean forests, woodlands and scrubs (south west & far south) Temperate grasslands, savannas and shrublands (east) Tropical and subtropical grasslands, savannahs and shrublands (north east) |
| Southern Australia | 1464 | Subtropical (west coast) Temperate Grassland | Mediterranean forests, woodlands and scrubs Temperate broadleaf and mixed forests Temperate grasslands, savannas and shrublands Montane grasslands and shrublands |

*Comment:*

5. The discussion's comparative analysis of the SPI and SPEI offers valuable methodological insights. However, stronger integration with region-specific climatic and biophysical drivers would benefit the interpretation. Additionally, the spatial specificity of distinctions between SPI and SPEI across sub-regions remains insufficiently delineated, limiting the granularity of conclusions.

*Response:*

This paper focussed on a broadscale analysis across Australia using NRM regions to delineate impacts. However, some of the insights may be scale dependent and analysis of smaller extents such as local government areas and basins may reveal a more locally relevant outcome. As discussed above, we have used the NRM regions to be consistent with recommended approaches for climate change assessment in Australia. We agree with the reviewer that the scale of the NRM regions is often insufficient to draw localised conclusions, which is why we have also provided regionalised drought characteristics for Australian Local Government Areas (566 sub-regions included) and River Basin (219 sub-regions included) as part of a supplementary dataset to this paper (https://doi.org/10.6084/m9.figshare.26343823). This delineated dataset may be used by readers to investigate localised impacts of the projected changes, which cannot all be included in this paper due to the number sub-regions involved. We also use the methodology presented within this paper as the basis from which to develop regionalised specific drought indices for a range of different region types (Local Government, Bio-Regions, NRM Regions, Regional Planning Areas, River Basins, and Disaster Districts) in Queensland, which are presented as a dashboard product through: (https://www.longpaddock.qld.gov.au/qld-future-climate/dashboard-cmip6/#responseTab5). As the reviewer suggests we will expand on our discussion of how meteorological droughts interact with biophysical factors, including land cover in section 4.3.

*Comment:*

Can more spatiotemporal visualizations (e.g., seasonal or interannual variability in drought indices in different regions) be incorporated to elucidate sub-regional heterogeneity clearly?

*Response:*

We will include more spatial visualisations of the 10$^{th}$ and 90$^{th}$ percentile of changes to the different drought characteristics along with the multi-model average. This will give a better understanding of the uncertainty of the projected changes and will better highlight regional differences.

The focus of our paper was on SPI-12 and SPEI-12 which includes the previous 12 months (annual) of accumulated of rainfall (and PET for SPEI), which is not suited to assessing seasonal variability. For this, a 3-month accumulation period would be better suited, which is broadly linked to agricultural droughts but outside the scope of our current work. We adopted a 12-month accumulation period for our assessments of SPI and SPEI as this was considered as a suitable timeframe for water deficits to impact various hydrological and agricultural systems (Zargar et al., 2011).

*Comment:*

The discussion should also explicitly articulate linkages between index disparities and potential localized environmental drivers, such as land cover status.

*Response:*

We thank the reviewer for their comment. We will include some discussion of how land cover change is incorporated into the projections (Eyring et al., 2016) and the associated impacts on meteorological droughts. We will also expand on our discussion of the interaction between meteorological droughts and environmental factors, including land cover as suggested by the reviewer in section 4.3.

**References:**

Chapman, S., Syktus, J., Trancoso, R., Thatcher, M., Toombs, N., Wong, K. K.-H., & Takbash, A. (2023). Evaluation of Dynamically Downscaled CMIP6-CCAM Models Over Australia. *Earth's Future*, *11*(11), e2023EF003548. https://doi.org/10.1029/2023EF003548

Chapman, S., Syktus, J., Trancoso, R., Toombs, N., & Eccles, R. (2024). Projected changes in mean climate and extremes from downscaled high-resolution CMIP6 simulations in Australia. *Weather and Climate Extremes*, *46*, 100733. https://doi.org/10.1016/j.wace.2024.100733

CSIRO and Bureau of Meteorology. (2015). *Climate Changein Australia Information for Australia's Natural Resource Management Regions: Technical Report*. CSIRO and Bureau of Meteorology.

Eyring, V., Bony, S., Meehl, G. A., Senior, C. A., Stevens, B., Stouffer, R. J., & Taylor, K. E. (2016). Overview of the Coupled Model Intercomparison Project Phase 6 (CMIP6) experimental design and organization. *Geoscientific Model Development*, *9*(5), 1937–1958. https://doi.org/10.5194/gmd-9-1937-2016

Grose, M. R., Narsey, S., Delage, F. P., Dowdy, A. J., Bador, M., Boschat, G., Chung, C., Kajtar, J. B., Rauniyar, S., Freund, M. B., Lyu, K., Rashid, H., Zhang, X., Wales, S., Trenham, C., Holbrook, N. J., Cowan, T., Alexander, L., Arblaster, J. M., & Power, S. (2020). Insights From CMIP6 for Australia's Future Climate. *Earth's Future*, *8*(5), e2019EF001469. https://doi.org/10.1029/2019EF001469

Kirono, D. G. C., Round, V., Heady, C., Chiew, F. H. S., & Osbrough, S. (2020). Drought projections for Australia: Updated results and analysis of model simulations. *Weather and Climate Extremes*, *30*, 100280. https://doi.org/10.1016/j.wace.2020.100280

Wasko, C., Guo, D., Ho, M., Nathan, R., & Vogel, E. (2023). Diverging projections for flood and rainfall frequency curves. *Journal of Hydrology*, *620*, 129403. https://doi.org/10.1016/j.jhydrol.2023.129403

Zargar, A., Sadiq, R., Naser, B., & Khan, F. I. (2011). A review of drought indices. *Environmental Reviews*, *19*(NA), 333–349. https://doi.org/10.1139/a11-013

---

## Author Response (AR1)

*General Comment:*

The paper describes a set of drought projections for Australia developed by dynamically downscaling CMIP6 global climate models. A possible range of future drought conditions is considered that span multiple emissions pathways and model configurations. Future drought conditions are described through event frequency, duration, spatial extent and time spent in drought, in terms of changes in two commonly used drought metrics. The paper is well-structured, the information is clearly presented and the key results appropriately discussed. I recommend publication after minor revisions, which I describe below.

*Response:*

We thank the reviewer for the time spent on our manuscript and for the positive and constructive comments provided. Our comments below indicate where we have made changes to the manuscript to address these concerns.

*Comment:*

One key issue is the use of SPEI. This metric, being the difference between precipitation and potential evapotranspiration, is intended to reflect the atmospheric water balance and thereby give a complementary view to SPI-based drought. However, the use of potential evapotranspiration in the calculation of SPEI makes SPEI unrealistic in many water-limited parts of Australia, where actual evapotranspiration does not approach the potential upper limit. So, any projected worsening of PET-related conditions is merely an indication of an increase in atmospheric demand for moisture, rather than a conclusive reduction in water stores. I suggest this issue is more adequately discussed in the paper, including the implications in the interpretation of SPEI-based projections of drought.

*Response:*

In accordance with the reviewer's suggestion, we have strengthened the discussion of the SPEI-based drought projections. Specifically, we have added the following in our discussion of the differences between SPI and SPEI (refer to page 22 of our revised manuscript for implementation):

*"However, further PET increases which drive SPEI in water-limited regions (Rangelands and Southern Australia) are unlikely to have as much consequence as in humid regions where the potential upper limit of actual evaporation has not already been met."*

We further highlight the potential limitation of SPEI in water limited regions (refer to page 23 of our revised manuscript for implementation):

*"On the other hand, there is potential that the SPEI could overestimates future drought magnitudes, **especially in water-limited regions** and may better represent a conservative upper limit of potential future drought risk."*

Additionally, as suggested by the reviewer we have expanded on our discussion of the implications (refer to page 23 of our revised manuscript for implementation):

*"It should be noted that increases to SPEI may not necessarily translate into on the ground changes, especially in water-limited environments where PET is already far greater than*

*precipitation. In these regions, which includes most of Australia the timing and magnitude of precipitation may be a more important consideration, and as such care must be taken when interpreting the SPEI-based drought projections."*

*Comment:*
The second key issue is the lack of attention given to the uncertainty of the projections. While using a multi-model ensemble and multiple emissions pathways goes some way to addressing uncertainty, the drought projections should be presented along with quantified uncertainty estimates. Moreover, the issue of uncertainty propagation from GCM through to downscaling technique to RCM was not addressed.

*Response:*
We thank the reviewer for their comment. We had attempted to show the uncertainty from the multi-model ensemble and multiple emissions pathways using timeseries plots of all ensemble members (Fig. 3 and Fig. 4), probability density function plots (Fig. 5 and Fig. 6 and Fig. 8 and Fig. 9), and boxplots (Fig. 11). In accordance with the reviewer's comment, we have updated our spatial maps (Fig. 10 and Fig. S15) to include hatching based on the quantitative signal-to-noise ratio to determine where the climate change signal emerges from uncertainty of the projections. We have updated the caption for this figure and the following text has been added to the methodology to introduce the approach (refer to page 8 of our revised manuscript for implementation):

*"To determine where there is confidence in the changes to the drought metrics, we adopt the signal-to-noise ratio to see where the climate change signal emerges over the 'noise' of the model ensemble (Hawkins et al., 2014). Here, the model uncertainty is considered as noise using the standard deviation of the projections (Hawkins and Sutton, 2011). We calculate the signal from the 11-model average, while the noise is derived from the standard deviation of all 15 projections (Chapman et al., 2024). Stippling is shown on the ensemble mean and median change maps where the signal-to-noise ratio is greater than 1.0 (Chapman et al., 2024; Hawkins et al., 2014; Hawkins & Sutton, 2011)."*

Additionally, we have added figures to the supplementary materials, showing spatial maps of the 10[th] and 90[th] percentile of changes along with the multi-model average (Fig. S16 to Fig. S20). We have also included an additional figure of the changes to precipitation and PET from CCAM in the supplementary materials (Fig. S2). The results section of our manuscript has been updated to reflect these changes (refer to pages 18-19 of our revised manuscript for implementation):

*"For the percent time in drought, frequency, and duration of extreme droughts, there were few regions where the signal-to-noise ratio was greater than one for SPI (Fig. 10). Significant increases can be noted in south-west Western Australia, in southern Victoria, southern South Australia and in western Tasmania under the high emissions scenario (SSP370), which are seen to reflect the spatial changes of mean precipitation (Fig. S2). In southwest Western Australia, SPI related extreme droughts were projected to occur both more frequently and last longer, leading to considerable increases in the percent time in drought. By contrast, the increases to the percent time in drought in southern Victoria, southern South Australia and in western Tasmania appears to be principally the result of increased drought frequency, with less clear*

*changes noted for drought duration. In addition to these regions, there were also significant increases to the percent time in moderate to extreme drought for the Gulf of Carpentaria and Northeastern Queensland for SSP370 by the 2090s (Fig. S15), which was not evident in the extreme droughts. For the remainder of the country, the results of SPI tended to be more uncertain. Interestingly, there were no regions of Australia where there was a significant reduction to the time spent in extreme drought.*

*For SPEI, there was wide model agreement for more frequent and longer drought events for the majority of the continent, particularly under SSP370 and for the end of the century (Fig. 10). This was especially true for the percentage time in drought, which is the result of both increasing drought frequency and duration. For parts of Northern Australia and Eastern Australia, there was generally less model agreement from the signal-to-noise ratio (as shown by the hatching) and the magnitude of the changes were typically smaller when compared to southern regions and the interior of the continent. Significant differences were noted between the 10th and 90th percentiles of projected changes to both SPI and SPEI, highlighting the uncertainty in these projections (Fig. S16 to Fig. S21)."*

We have added the following text to the methodology to highlight how GCMs compare to the RCM used in this study (refer to page 5 of our revised manuscript for implementation):

*"In the future, the climate change signal of the host GCMs from downscaling was shown to generally be preserved for precipitation, though with some differences in magnitudes in some regions, particularly in summer. For temperature changes, the downscaled models were shown to have good agreement with the host models across Australia (Chapman et al., 2024)."* (p. 5)

We note that PET was derived offline from CCAM using Penman-Monteith reference crop approach using CCAM outputs of solar radiation, vapour pressure, maximum and minimum temperature, mean sea level pressure, and wind speed. As such, we are unable to compare PET projections from CCAM to the GCMs and unable to compare drought metrics derived from the GCMs to CCAM.

*Comment:*

The final key issue is that one of the most crucial findings of the study needs to be made more prominent. The results show that more time is projected to be spent under extreme conditions, both wet and dry, and less time under 'normal' conditions, for some parts of Australia (Table 3). This result should be made more prominent, for example by featuring in the abstract. This result is important because it suggests that the combination of projected changes in the climate system is shifting the dial towards more extreme climatic conditions and motivates future research in understanding the physical processes responsible for the shift.

*Response:*

We thank the reviewer for pointing this out. In accordance with their suggestion, we have added these key findings into the abstract (refer to page 1 of our revised manuscript for implementation):

*"Increases to drought appear to have mostly come at the expense of 'normal' climatic conditions, with similar or increased time spent under extreme wet conditions, indicating an overall shift towards more extreme climatic conditions."*

And the conclusion (refer to page 25 of our revised manuscript for implementation):
*"These increases appear to have largely come at the expense of 'normal' climatic conditions, with little changes or small increases to time spent under extreme wet conditions, pointing towards an overall shift towards more extreme climatic conditions across Australia."*

Furthermore, we have expanded on our discussion of future drought by adding the following section in bold (refer to page 22 of our revised manuscript for implementation):
*"Interestingly, the increase in extreme droughts did not lead to a decrease in extreme wetness, but rather mostly reduced time in near normal climate conditions (Table 3).* **Indeed, in some regions there was an increase to the time spent in extreme wet conditions in the future, indicating an overall shift towards more extreme climatic conditions.** *This was due to a shift in the mean and an overall flattening of the PDFs of SPI and SPEI as seen in Fig. 5 and Fig. 6, leading to more time in drought conditions. Similar PDFs changes have been noted in global assessments of soil moisture, runoff, and the Palmer drought index under CMIP5 and CMIP6 (Zhao and Dai, 2015, 2022)."*

*Minor Comment:*
L84: this is a bit of a throw away line. I suggest turning this around by stating that since RCMs have been shown to estimate regional rainfall features with higher precision than GCMs, RCMs are more appropriate to study drought on the regional scale.
*Response:*
In accordance with the reviewer's suggestion, we have changed this line to (refer to page 3 of our revised manuscript for implementation):
*"However, while research to date has largely focussed on applying coarse GCM outputs to assess future droughts, RCMs have been shown to have more skill in representing key rainfall features and may therefore be better suited to study droughts at regional scales."*

*Minor Comment:*
Inter-model variability (Figure 11) is shown to be higher for SPEI and some drought characteristics. Can an explanation be offered for why this is? What are the implications of this variability on the interpretation of future drought changes?
*Response:*
We believe the variability of the projected changes is related to the mean projected change (i.e. the range of changes approximately scales with the mean change). We have therefore added the following into the discussion (refer to pages 20-21 of our revised manuscript for implementation):
 *"The inter-model variability appears to approximately scale with the mean change in the projections, indicating greater uncertainty for larger changes."*
*"While the sign of the change is clear in these regions, especially for SPEI, there is considerable inter-model variability in the magnitude of the projected changes (Fig. 11), which may necessitate decision makers to adopt an adaptive approach to planning for these future eventualities."*

*General Comment:*

The manuscript 'Meteorological Drought Projections for Australia from Downscaled high-resolution CMIP6 climate simulations' presents the future drought features (SPI and SPEI) based on the downscaled precipitation and potential evapotranspiration data. The work is well-presented. However, there are some issues that need to be clarified further before the publication.

*Response:*

We thank the reviewer for their time and constructive comments on our manuscript. Our comments below show where we have made changes to the manuscript to address these concerns.

*Comment:*

1. This study utilizes various drought characteristics, including duration, frequency, percent time (Figure 2), and shifts in the moving average, to predict future droughts. However, since the downscaling is applied only spatially, all temporal analyses could be conducted using GCM data. Yet, only Figure 10 presents a spatial map. What is the rationale for using downscaled data in this context?

*Response:*

We thank the reviewer for their comment. It is important to note that downscaling does improve the temporal and the spatial resolution of the projections (for instance CCAM has sub-daily data available). However, as this analysis was conducted using accumulated monthly precipitation and PET, outputs from GCMs could also be applied as suggested by the reviewer, though at a much coarser spatial resolution. We have found in previous work that downscaling improves the representation of precipitation and temperature, even when assessed at coarse spatial scales (Chapman et al., 2023), which we better highlight in our revised methodology (refer to page 5 of our revised manuscript for implementation):

*"The downscaling approach adopted has been shown to significantly improve the performance over the host GCMs for precipitation and temperature in all seasons, with the largest improvements noted for climate extremes, even when assessed across the four Australian IPCC regions (Chapman et al., 2023), which are similar to the NRM super-clusters adopted in this study. Across Australia as a whole, seasonal precipitation was shown to improve in all models, with an ensemble average improvement of 43% using the Kling-Gupta Efficiency, while the annual cycle of precipitation improved in most models with an ensemble average improvement of 13% (Chapman et al., 2023). Downscaling also improved the fraction of dry days, reducing the bias for too many low-rain days. These improvements have clear beneficial effects for the simulation of future droughts."*

The data visualization of such a complex analysis involving multiple sources of variation (i.e., emissions scenarios, time horizons, drought characteristics, drought severities and regions) is challenging and maps may not be the best type of graphic to convey the findings and communicate nuances under space constraints. For instance, we made a conscious choice in

the manuscript to combine the spatial maps into subplots where possible to allow for an easy comparison of changes to drought characteristics from both SPI and SPEI. For instance, Fig. 10 highlighted by the reviewer contains subplots of 36 maps for extreme droughts. Additional maps of changes to moderate droughts can be seen in the supplementary materials (Fig. S14). We have added additional spatial maps of the 10th and 90th percentile changes to drought metrics to better understand the spatial uncertainty of the projections (Fig. S16 to Fig. S20), as well as maps of changes to precipitation and PET (Fig. S2).

We also wanted to highlight that one of the outputs of this contribution is the dataset of regionalised drought characteristics for Australian Local Government Areas and River Basin (https://doi.org/10.6084/m9.figshare.26343823), which is only possible due to the finer granularity of the downscaled projections. We better highlight this advantage in the methodology (changes in bold; refer to pages 8-9 of our revised manuscript for implementation):

*"Additional supplementary datasets tailoring projected drought impacts to Australian Local Government Areas (566 sub-regions included) and River Basins (219 sub-regions included) are also made available (Eccles, 2024), thanks to the high-resolution projections used in this study."*

*Comment:*
2. Why did the author choose to use downscaled data from the Conformal Cubic Atmospheric Model (CCAM)? What advantages does CCAM offer compared to other downscaled datasets? Additionally, how can you demonstrate that drought characteristics derived from the downscaled data are more reliable or accurate than those based on raw GCM data?
*Response:*
The reviewer is correct that there are other downscaled datasets available as part of the CORDEX CMIP6 experiment. However, at the time that this work was undertaken, only the CCAM dataset was available for analysis. This downscaled dataset is also advantageous over other datasets, as it is the largest ensemble available (15 model runs per emission scenario) and run at the highest resolution (10 km). We have added the following text to better highlight this advantage (refer to page 5 of our revised manuscript for implementation):
*"This represents the largest downscaled ensemble of projections in Australia ran at the highest resolution."*

We adopted reference crop evapotranspiration (PET) for calculating the SPEI, which was derived offline from CCAM requiring daily data for solar radiation, vapour pressure, maximum and minimum temperature, mean sea level pressure, and wind speed. This method for deriving PET is more data intensive and complex than alternatives but provides better estimations of PET compared to pan evaporation or simple temperature-based PET estimations. The approach is not available from other downscaled ensembles or from the raw GCM data. We have added the following to the introduction to better elucidate the advantages of this approach (refer to page 7 of our revised manuscript for implementation):
*"This method for deriving PET is more intensive than simpler temperature-based approaches but is recommended where data is available (Beguería et al., 2014; Hosseinzadehtalaei et al., 2017; Sheffield et al., 2012)."*

As we have derived our PET offline to CCAM no direct comparison to the host GCMs is possible. We have however, previously compared how downscaling improves the simulation of other variables such as precipitation and temperature, which will have clear benefits for the simulation of droughts. In line with the reviewer's comment, we have added the following to the introduction, which we also highlighted in response to comment 1 (refer to page 5 of our revised manuscript for implementation):

*"Across Australia as a whole, seasonal precipitation was shown to improve in all models, with an ensemble average improvement of 43% using the Kling-Gupta Efficiency, while the annual cycle of precipitation improved in most models with an ensemble average improvement of 13% (Chapman et al., 2023). Downscaling also improved the fraction of dry days, reducing the bias for too many low-rain days. These improvements have clear beneficial effects for the simulation of future droughts."*

*Comment:*
3. Is there any result about the comparison between the downscaled data and original data (such as precipitation and potential evapotranspiration) to evaluate the downscaling methods' performance?

*Response:*
As we note above, we have evaluated the added value of downscaling on precipitation and temperature, and undertaken comparisons with host models for historical (Chapman et al., 2023) and future (Chapman et al., 2024) projections, which we now highlight in the methodology (refer to page 5 of our revised manuscript for implementation):

*"The downscaling approach adopted was shown to significantly improve the performance over the host GCMs for precipitation and temperature in all seasons, with the largest improvements noted for climate extremes, even when assessed across the four Australian IPCC regions (Chapman et al., 2023), which are broadly similar to the NRM super-clusters adopted in this study. Across Australia as a whole, seasonal precipitation was shown to improve in all models, with an ensemble average improvement of 43% using the Kling-Gupta Efficiency, while the annual cycle of precipitation improved in most models with an ensemble average improvement of 13%. These improvements have clear beneficial effects for the simulation of future droughts. Downscaling also improved the fraction of dry days, reducing the bias for too many low-rain days. In the future, the climate change signal of the host GCMs from downscaling was shown to generally be preserved for precipitation, though with some differences in magnitudes in some regions, particularly in summer. For temperature changes, the downscaled models were shown to have good agreement with the host models across Australia (Chapman et al., 2024)."*

As also noted above, PET was derived offline from the model, and so we could not compare the performance from CCAM and the host GCMs.

*Comment:*
4. The study area was divided into four distinct regions—Eastern Australia, Northern Australia, the Rangelands, and Southern Australia—based on climatic and biophysical characteristics. However, the specific climatic and biophysical parameters used for this classification were not

explicitly defined. Including detailed information on climate patterns (e.g., precipitation regimes, seasonal variations), dominant vegetation types, and temperature ranges could enhance the clarity of the classification framework. Such specifications would facilitate a more comprehensive interpretation of the analytical results by providing critical contextual information about regional environmental variations.

*Response:*

We thank the reviewer for pointing out the lack of information regarding how the NRM regions were defined. It is important to note that we did not classify these regions ourselves. Rather, we adopt pre-defined regions classified by Australian Federal agencies to specifically assess climate change in Australia (CSIRO and Bureau of Meteorology, 2015). In accordance with the reviewer's recommendation, we have revised Fig. 1 in the revised manuscript to include the major climate regions as a background and have added a table in the supplementary materials which details the dominant climate and ecological characteristics within each of the super-clusters using information from (CSIRO and Bureau of Meteorology, 2015). We have added the following sentence to the methodology to reflect this change (refer to page 4 of our revised manuscript for implementation):

*"Details of the dominant climate zones and ecological characteristics within each of these super clusters are presented in Table S1."*

*Comment:*

5. The discussion's comparative analysis of the SPI and SPEI offers valuable methodological insights. However, stronger integration with region-specific climatic and biophysical drivers would benefit the interpretation. Additionally, the spatial specificity of distinctions between SPI and SPEI across sub-regions remains insufficiently delineated, limiting the granularity of conclusions.

The discussion should also explicitly articulate linkages between index disparities and potential localized environmental drivers, such as land cover status.

*Response:*

As the reviewer suggests we have expanded our discussion of how meteorological droughts interact with biophysical factors, including land cover in section 4.3 (refer to page 24 of our revised manuscript for implementation):

*"Both positive and negative changes in landcover can influence meteorological droughts through changes in precipitation, temperature, and windspeed (due to changing surface roughness), which influence both SPI and SPEI. For instance, in southwest Western Australia largescale anthropogenic landcover changes were shown to partially drive long-term declines in precipitation along coastal regions and increases in inland regions (Pitman et al., 2004; Timbal and Arblaster, 2006). Further landcover changes as a result of climate change or other anthropogenic activities may therefore work to further exacerbate or mitigate future droughts depending on the region and the changes. The projections included in this study include landcover changes which are prescribed according to the emissions scenario (Eyring et al., 2016). These changes are, however, not dynamic or responsive to changes in the climate and as such could respond differently in the future, potentially impacting on the magnitude of the drought changes presented."*

We assess NRM regions in the paper to enable the results to be interpretable (more regions require more subplots) and as NRM regions have commonly been used to assess climate change impacts in Australia. We have made available much more granular data as suggested by the reviewer which may be applied for these local-scale analyses. We have added the following text in our discussion to better highlight this fine scale dataset (refer to page 21 of our revised manuscript for implementation):

*"We provide supplementary datasets tailoring these projections to Australian River Basins and Local Government Areas (Eccles, 2024). These datasets provide derived drought metrics at a much more granular scale, which may be useful for informing local and regional scale decisions on adaptation and drought preparedness."*

*Comment:*
Can more spatiotemporal visualizations (e.g., seasonal or interannual variability in drought indices in different regions) be incorporated to elucidate sub-regional heterogeneity clearly?
*Response:*
In line with the reviewer's comment, we have included plots in the supplementary materials showing the interannual variability of projected droughts in each of the regions (Fig. S25 to Fig. S48). As each model has a different sequence of wet and dry events, we show all models so that the interannual variability of the projections is evident. We have added the following to the results section to reflect these changes (refer to page 10 of our revised manuscript for implementation):

*"Interannual variability from the different projections in each of the regions are presented in Fig. S25 to Fig. S48."*

The focus of our paper was on SPI-12 and SPEI-12 which includes the previous 12 months of accumulated of rainfall (and PET for SPEI), which is not suited to assessing seasonal variability. For this, a 3-month accumulation period would be better suited, which is broadly linked to agricultural droughts but outside the scope of this paper. We adopted a 12-month accumulation period for our assessments of SPI and SPEI as this was considered as a suitable timeframe for water deficits to impact various hydrological and agricultural systems (Zargar et al., 2011).

**References:**

Beguería, S., Vicente-Serrano, S. M., Reig, F., & Latorre, B. (2014). Standardized precipitation evapotranspiration index (SPEI) revisited: Parameter fitting, evapotranspiration models, tools, datasets and drought monitoring. *International Journal of Climatology*, *34*(10), 3001–3023.

Chapman, S., Syktus, J., Trancoso, R., Thatcher, M., Toombs, N., Wong, K. K.-H., & Takbash, A. (2023). Evaluation of Dynamically Downscaled CMIP6-CCAM Models Over Australia. *Earth's Future*, *11*(11), e2023EF003548. https://doi.org/10.1029/2023EF003548

Chapman, S., Syktus, J., Trancoso, R., Toombs, N., & Eccles, R. (2024). *Projected Changes in Mean Climate and Extremes from Downscaled High-Resolution Cmip6 Simulations in Australia* (SSRN Scholarly Paper No. 4836517). https://doi.org/10.2139/ssrn.4836517

CSIRO and Bureau of Meteorology. (2015). *Climate Changein Australia Information for Australia's Natural Resource Management Regions: Technical Report*. CSIRO and Bureau of Meteorology.

Eyring, V., Bony, S., Meehl, G. A., Senior, C. A., Stevens, B., Stouffer, R. J., & Taylor, K. E. (2016). Overview of the Coupled Model Intercomparison Project Phase 6 (CMIP6) experimental design and organization. *Geoscientific Model Development*, *9*(5), 1937–1958. https://doi.org/10.5194/gmd-9-1937-2016

Hawkins, E., Anderson, B., Diffenbaugh, N., Mahlstein, I., Betts, R., Hegerl, G., Joshi, M., Knutti, R., McNeall, D., Solomon, S., Sutton, R., Syktus, J., & Vecchi, G. (2014). Uncertainties in the timing of unprecedented climates. *Nature*, *511*(7507), Article 7507. https://doi.org/10.1038/nature13523

Hawkins, E., & Sutton, R. (2011). The potential to narrow uncertainty in projections of regional precipitation change. *Climate Dynamics*, *37*(1), 407–418. https://doi.org/10.1007/s00382-010-0810-6

Hosseinzadehtalaei, P., Tabari, H., & Willems, P. (2017). Quantification of uncertainty in reference evapotranspiration climate change signals in Belgium. *Hydrology Research*, *48*(5), 1391–1401. https://doi.org/10.2166/nh.2016.243

Pitman, A. J., Narisma, G. T., Pielke Sr., R. A., & Holbrook, N. J. (2004). Impact of land cover change on the climate of southwest Western Australia. *Journal of Geophysical Research: Atmospheres*, *109*(D18). https://doi.org/10.1029/2003JD004347

Sheffield, J., Wood, E. F., & Roderick, M. L. (2012). Little change in global drought over the past 60 years. *Nature*, *491*(7424), Article 7424. https://doi.org/10.1038/nature11575

Timbal, B., & Arblaster, J. M. (2006). Land cover change as an additional forcing to explain the rainfall decline in the south west of Australia. *Geophysical Research Letters*, *33*(7). https://doi.org/10.1029/2005GL025361

Zargar, A., Sadiq, R., Naser, B., & Khan, F. I. (2011). A review of drought indices. *Environmental Reviews*, *19*(NA), 333–349. https://doi.org/10.1139/a11-013

---

## Referee Report (RR1)

Review : Eccles et al., 2025

**High-resolution downscaled CMIP6 drought projections for Australia**

**General comments**

This paper "High-resolution downscaled CMIP6 drought projections for Australia" is well written, with a clear structure, well-chosen illustrations and well explained results to address their research question: representing the characteristics of meteorological droughts in Australia at high-resolution.

I particularly commend the introduction and the discussion, giving a very good perspective on the study and its results.

I feel the comments from the previous reviewers have been well addressed, with pertinent elements added to the article and comprehensive answers written.

I recommend this article for publication and only have a few minor comments, notably to improve the presentation of the methodology and the choices made in the downscaling part of the study.

**Minor comments**

My main issue in the paper is related to the section "2.2 Data", which I feel could be made clearer. It would be helpful to better present the project for which the downscaling was done, to explain some of the choices made. Why "some model variants were downscaled multiple times" (l.114) ? This sentence seems to come too early, and corresponds to the explanation given l. 124 and Table 1? How did the authors select their model ensemble? Why are some CMIP6 models downscaled with different set-ups and not others?
This needs to be better explained, especially since in the later sections, we need to understand when it is pertinent to analyse the 15-model ensemble, or rather consider an 11-model average.
It would made the section 2.4 clearer (l. 196 to 204).

Also, the authors mention that the downscaling approach significantly improve the performance on temperature and precipitation than using the coarse scale GCMs. Against which observation data? Since we are in the data section and that observation data are directly mentioned in the following paragraph, I feel this information could be added.

Still in the methodological part, section 2.3, the authors mention a calibration on the historical period to fit the SPI and SPEI distribution in the future period.
I do not understand fully how this was conducted, and also how it fits with the analysis later on of the changes in the SPI and SPEI distributions and shifts in percentiles, if these distributions have been calibrated…
The paragraph from l. 174 to l. 180 needs to be made clearer, with more details given on the transfer function used for the calibration, if there was a calibration. The first section of the results (3.1) seems to imply that there was none, since you look at the biases between observation and simulation. Otherwise, how does the calibration changes the performances? I am confused.

**Specific comments for the authors**

Introduction

- There is a redundant information in the introduction, with l.65 to 69 similar to l.84 to 86. Since the structure of your introduction shows: the use of GCM (l.54 to 64), their limits and the use of RCMs (l.65 to 70), but last studies used for CMIP5 (l.71 to 81), I feel you could

reorganise slightly l81 to 86. It would avoid the redundancy and better highlight that you want to work with CMIP6, and it is the main novelty of your study.

- To help the method section of the article, I think you could mention the RCM you used in the end of your introduction and detail a bit more the Queensland Future Climate Science Program (QFCSP).

**Method**

- P.7: you use a notation SPI/SPEI. If I understood correctly, it means "SPI or SPEI". Be careful, it looks like a ratio. I feel this need to be changed.
- Sentence l. 212-213 is not clear. A percentile can not be a shift. The sentence is not correct. Ta define a shift, you need a reference. So I think I understand what you mean. If the 50$^{th}$ percentile in the projected distribution matches the 40$^{th}$ percentile in the historical period, therefore there is a 10% shift towards dryness.

**Results**

I feel the results are very well explained with pertinent illustration. Still I have a few comments.

- Section 3.2.2: how do you define the "area affected by droughts"? Is it all pixels for which there is at least one drought event (depending on the severity of the drought event defined)?
- L. 352-354: unclear : "differences between the 10$^{th}$ and 90$^{th}$ percentiles"?

**Discussion**

- L. 383: "time in" ?
- L. 424: What does "this" refer to? The elevated PET?
- L. 454: "may better" → "would rather" ?

---

## Referee Report (RR2)

Review : Eccles et al., 2025

**High-resolution downscaled CMIP6 drought projections for Australia**

The revised manuscript is still very well written. The comments to improve slightly the structure and the presentation of the methodology have been very well addressed. The authors gave relevant responses to each comment and improved a few sections of the article accordingly. The downscaling method used to select the climate model ensembles and the valorisation of the climate outputs to study the drought indices are well explained and a lot clearer than in the previous version.

I believe this paper well addresses its research question, with now a clear methodology and comprehensive analyses of the future characteristics of drought in Australia and recommend it for publication.

**A few specific comments**

l.179 and 185: Some « SPI/SPEI » left.

L85-86: Issue in the sentence: "The downscaling was performed using dynamically downscaled using the Conformal Cubic Atmospheric Model (CCAM)"

---

## Author Response (AR2)

Point by Point Response for Reviewer 1

*General Comment:*

This paper "High-resolution downscaled CMIP6 drought projections for Australia" is well written, with a clear structure, well-chosen illustrations and well explained results to address their research question: representing the characteristics of meteorological droughts in Australia at high-resolution.

I particularly commend the introduction and the discussion, giving a very good perspective on the study and its results.

I feel the comments from the previous reviewers have been well addressed, with pertinent elements added to the article and comprehensive answers written.

I recommend this article for publication and only have a few minor comments, notably to improve the presentation of the methodology and the choices made in the downscaling part of the study.

*Response:*

*Thank you for the time spent reviewing the article and for the positive and constructive feedback. Our comments below indicate where we have made changes to the manuscript to address these concerns.*

*Minor Comment:*

My main issue in the paper is related to the section "2.2 Data", which I feel could be made clearer. It would be helpful to better present the project for which the downscaling was done, to explain some of the choices made. Why "some model variants were downscaled multiple times" (l.114) ? This sentence seems to come too early, and corresponds to the explanation given l. 124 and Table 1? How did the authors select their model ensemble? Why are some CMIP6 models downscaled with different set-ups and not others? This needs to be better explained, especially since in the later sections, we need to understand when it is pertinent to analyse the 15-model ensemble, or rather consider an 11-model average. It would made the section 2.4 clearer (l. 196 to 204).

*Response:*

We have revised our 'Data' Section to make the downscaling approach clearer by including additional information on the downscaling methodology and the model selection process. The model selection process was based on a combination of data availability, model performance, model independence, and the climate change signal. Both atmosphere only and ocean-coupled simulations were considered to test improvements from new ocean coupling modelling techniques. These choices were made with all climate hazards in mind and not just droughts. We have explained the downscaling and model selection approach in our manuscript (refer to lines 115-143 of our revised manuscript for implementation of these changes):

*'We used the CCAM model developed by CSIRO (McGregor & Dix, 2008) to dynamically downscale 15 CMIP6 GCMs. Typically, dynamical downscaling involves running an RCM over a limited domain, with the host GCM forcing the lateral boundaries. CCAM differs as it is a global stretched grid model and so is run for the entire globe, with the domain of interest run at a higher resolution. Here, instead of providing lateral boundaries, the regional atmosphere in CCAM is influenced by large scale climate simulated from the host GCM, while at a small scale*

*the atmosphere is allowed to evolve freely (Thatcher & McGregor, 2009). CCAM was run using a stretched C288 grid in both atmospheric and ocean-coupled versions, which consists of a model resolution of approximately 10 km. In total, 35 vertical layers in the atmosphere and 30 layers in the ocean for the ocean-coupled models were applied (Thatcher et al., 2015). A downscaling approach outlined by Hoffman et al. (2016) was used, which involved bias correcting the sea surface temperatures and sea ice from the host GCMs prior to downscaling. This approach has been found to improve the simulations of climate from CCAM and other regional climate models (Hoffmann et al., 2016; Kim et al., 2020; Lim et al., 2019).*

*We used an ensemble of 60 downscaled climate model simulations derived from 11 different CMIP6 GCMs (Table 1). The ensemble consists of 15 runs for historical simulations and three sets of 15 runs for future simulations under three Shared Socioeconomic Pathways (SSP126, SSP245 and SSP370), representing low, moderate, and high-emissions pathways, respectively. The ensemble of GCMs used in this study was selected in order to best represent the future spread in the climate change signal from the ensemble of global CMIP6 models, while prioritising models which were better able to represent the Australian climate (Trancoso et al., 2023). For instance, we selected several GCMs spread across the distribution of projected temperature and precipitation changes, but also outlier models representing the driest (ACCESS-ESM1.5) and wettest (EC-Earth3) GCMs (Chapman et al., 2023). All the GCMs were assessed based on their ability to represent Australia's precipitation and temperature compared to Australian Gridded Climate Data Project (AGCD; Evans et al., 2020) observational data between 1995 and 2014 using the Kling-Gupta Efficiency (KGE). The climate change signal at the mid and end of the century was evaluated and combined with the KGE score from the historical simulations to select the best performing ensemble runs from the different GCMs through a Skill-Spread-Selection algorithm (Trancoso et al., 2023). Five of the CCAM simulations were run using dynamic atmosphere-ocean coupling as presented in* **Error! Reference source not found.** *in order to better understand the influence of ocean coupling on model outputs. Additionally, three variants including the best performing, the wettest, and the driest ensemble member from the large ensemble (40 members) of ACCESS-ESM1.5 simulations were considered, to facilitate assessments of intra-model variability and the influence of initial conditions. This represents the largest downscaled ensemble of projections in Australia ran at the highest resolution.'*

*Minor Comment:*
Also, the authors mention that the downscaling approach significantly improve the performance on temperature and precipitation than using the coarse scale GCMs. Against which observation data? Since we are in the data section and that observation data are directly mentioned in the following paragraph, I feel this information could be added.
*Response:*
We have clarified which observational dataset was used to determine the improvements in downscaling which are detailed in (Chapman et al., 2023) (refer to lines 147-150 of our revised manuscript for implementation):
*'The downscaling approach adopted has been shown to significantly improve the performance over the host GCMs for precipitation and temperature in all seasons when compared to gridded AGCD observational data, with the largest improvements noted for climate extremes, even when assessed across the four Australian IPCC regions (Chapman et al., 2023), which are similar*

*to the NRM super-clusters adopted in this study.'*

*Minor Comment:*
Still in the methodological part, section 2.3, the authors mention a calibration on the historical period to fit the SPI and SPEI distribution in the future period.
I do not understand fully how this was conducted, and also how it fits with the analysis later on of the changes in the SPI and SPEI distributions and shifts in percentiles, if these distributions have been calibrated…
The paragraph from l. 174 to l. 180 needs to be made clearer, with more details given on the transfer function used for the calibration, if there was a calibration. The first section of the results (3.1) seems to imply that there was none, since you look at the biases between observation and simulation. Otherwise, how does the calibration changes the performances? I am confused.

*Response:*
The term calibration period relates to the period of data used to fit the Gamma and Log-Logistic distributions to derive SPI and SPEI, respectively. The fitted parameter values are then applied to estimate the SPI and SPEI values from the precipitation and potential-evapotranspiration data. There was no 'calibration' performed in our analysis. To avoid this confusion, we have changed all references from 'calibration period' to 'historical period' throughout the manuscript. Additionally, we have revised the description of text which was confusing (refer to lines 192-198 of our revised manuscript for implementation):

*'However, when assessing changes to these indices as a result of climate change, a historical period is commonly adopted to fit the distribution. The fitted distribution parameter values are then applied to estimate the SPI and SPEI for the future period, allowing for a comparison of projected future dryness and wetness compared to the recent past. For our assessment, we have adopted a historical period from 1981-2010 to fit the Gamma and Log-Logistic distributions for SPI and SPEI, respectively. Fitted distribution values were then used to calculate SPI and SPEI over the full timeseries, containing both historical and future simulations (1981-2100).'*

*Specific Comment:*
There is a redundant information in the introduction, with l.65 to 69 similar to l.84 to 86. Since the structure of your introduction shows: the use of GCM (l.54 to 64), their limits and the use of RCMs (l.65 to 70), but last studies used for CMIP5 (l.71 to 81), I feel you could reorganise slightly l81 to 86. It would avoid the redundancy and better highlight that you want to work with CMIP6, and it is the main novelty of your study.

*Response:*
Thank you for bringing this to our attention. We have revised the introduction by reworking what was said previously in lines 81-86 into the earlier paragraphs as suggested.

*Specific Comment:*
To help the method section of the article, I think you could mention the RCM you used in the

end of your introduction and detail a bit more the Queensland Future Climate Science Program (QFCSP).

*Response:*

In accordance with the reviewer's recommendation we have revised the end of the introduction to provide some further details of the downscaling and of the QFSCP (refer to lines 84-93 of our revised manuscript for implementation):

*'This study expands on the available body of knowledge for future meteorological droughts in Australia, employing an ensemble of 60 high-resolution dynamically downscaled CMIP6 simulations (15 historical and 45 future simulations). The downscaling was performed using dynamically downscaled using the Conformal Cubic Atmospheric Model (CCAM), and followed the CORDEX experimental protocol. These projections form part of the Queensland Future Climate Science Program (QFCSP) and are available at a 10 km resolution over the Australian continent as the QldFCP-2 data set (Queensland Future Climate Projections 2). The QldFCP-2 simulations were shown to lead to improvements in mean climate over the historical period, however, the largest improvements were noted for climate extremes, particularly over coastal and mountainous regions (Chapman et al., 2023). These projections form part of a national strategy for climate projections, contributing to a wider set of downscaled CORDEX compliant projections for Australia as part of the National Partnership for Climate Projections (Grose et al., 2023), which will underpin climate services and adaptation planning nationally.'*

*Specific Comment:*

P.7: you use a notation SPI/SPEI. If I understood correctly, it means "SPI or SPEI". Be careful, it looks like a ratio. I feel this need to be changed.

*Response:*

As suggested, we have changed this to 'SPI or SPEI'

*Specific Comment:*

Section 3.2.2: how do you define the "area affected by droughts"? Is it all pixels for which there is at least one drought event (depending on the severity of the drought event defined)?

*Response:*

The area affected by drought, is the percentage of the area (i.e., number of grid cells affected by drought divided by total number of grid cells within a given region) at any given timestep, which are categorised as in drought (depending on the severity). This results in a timeseries of the percentage of area in drought for a given region. We have clarified this in the methods section (refer to lines 205-208 of our revised manuscript for implementation):

*'Here, the frequency is defined as the total number of events recorded over a given time period, the duration is the average duration of recorded drought events (in months), the percent time in drought is the fraction of time droughts occur, and the spatial extent is the number of grid cells affected by each drought severity category divided by total number of grid cells within a given region for each timestep.'*

*Specific Comment:*

L. 352-354: unclear : "differences between the 10th and 90th percentiles"?

*Response:*
We have revised this to make it clear we are referring to the range between the 10th and 90th percentiles (refer to lines 370-372 of our revised manuscript for implementation):
*'There was a large range between the 10th and 90th percentile ensemble projections for both SPI and SPEI (Fig. S16 to Fig. S21), highlighting the uncertainty in these projections.'*

*Specific Comment:*
Sentence l. 212-213 is not clear. A percentile can not be a shift. The sentence is not correct. Ta define a shift, you need a reference. So I think I understand what you mean. If the 50th percentile in the projected distribution matches the 40th percentile in the historical period, therefore there is a 10% shift towards dryness.
*Response:*
Your interpretation is correct. We have revised this sentence as suggested (refer to lines 229-230 of our revised manuscript for implementation):
*'We therefore assessed when significant changes to the long-term average values occurred based on a 10% and 20% shift towards dryness compared to the historical period.'*

*Specific Comment:*
L. 383: "time in" ?
*Response:*
We have clarified this in the text by changing 'time in' to "percentage of time spent in drought'.

*Specific Comment:*
L. 424: What does "this" refer to? The elevated PET?
*Response:*
We have clarified the text by changing 'this' to 'atmospheric water demand'.

*Specific Comment:*
L. 454: "may better" ⮕ "would rather" ?
*Response:*
We have made this change.

References:

Chapman, S., Syktus, J., Trancoso, R., Thatcher, M., Toombs, N., Wong, K. K.-H., & Takbash, A. (2023). Evaluation of Dynamically Downscaled CMIP6-CCAM Models Over Australia. *Earth's Future*, *11*(11), e2023EF003548. https://doi.org/10.1029/2023EF003548

Evans, A., Jones, D., Smalley, R., & Lellyett, S. (2020). An enhanced gridded rainfall analysis scheme for Australia. *Australian Bureau of Meteorology: Melbourne, VIC, Australia*, *66*, 55–67.

Grose, M. R., Narsey, S., Trancoso, R., Mackallah, C., Delage, F., Dowdy, A., Di Virgilio, G., Watterson, I., Dobrohotoff, P., Rashid, H. A., Rauniyar, S., Henley, B., Thatcher, M., Syktus, J., Abramowitz, G., Evans, J. P., Su, C.-H., & Takbash, A. (2023). A CMIP6-based multi-model downscaling ensemble to underpin climate change services in Australia. *Climate Services*, *30*, 100368. https://doi.org/10.1016/j.cliser.2023.100368

Hoffmann, P., Katzfey, J. J., McGregor, J. L., & Thatcher, M. (2016). Bias and variance correction of sea surface temperatures used for dynamical downscaling. *Journal of Geophysical Research*, *121*(21), 12,877-12,890. https://doi.org/10.1002/2016JD025383

Kim, Y., Rocheta, E., Evans, J. P., & Sharma, A. (2020). Impact of bias correction of regional climate model boundary conditions on the simulation of precipitation extremes. *Climate Dynamics*, *55*(11–12), 3507–3526. https://doi.org/10.1007/S00382-020-05462-5/FIGURES/10

Lim, C. M., Yhang, Y. B., & Ham, S. (2019). Application of GCM Bias Correction to RCM Simulations of East Asian Winter Climate. *Atmosphere 2019, Vol. 10, Page 382*, *10*(7), 382. https://doi.org/10.3390/ATMOS10070382

McGregor, J. L., & Dix, M. R. (2008). An updated description of the conformal-cubic atmospheric model. In *High Resolution Numerical Modelling of the Atmosphere and Ocean* (pp. 51–75). Springer New York. https://link.springer.com/chapter/10.1007/978-0-387-49791-4_4

Thatcher, M., McGregor, J., Dix, M., & Katzfey, J. (2015). A new approach for coupled regional climate modeling using more than 10, 000 cores. *IFIP Advances in Information and Communication Technology*, *448*, 599–607. https://doi.org/10.1007/978-3-319-15994-2_61/COVER

Thatcher, M., & McGregor, J. L. (2009). Using a Scale-Selective Filter for Dynamical Downscaling with the Conformal Cubic Atmospheric Model. *Monthly Weather Review*, *137*(6), 1742–1752. https://doi.org/10.1175/2008MWR2599.1

Trancoso, R., Syktus, J., Toombs, N., & Chapman, S. (2023). *Assessing and selecting CMIP6 GCMs ensemble runs based on their ability to represent historical climate and future climate change signal* (Nos. EGU23-11412). EGU23. Copernicus Meetings. https://doi.org/10.5194/egusphere-egu23-11412

---

## Author Response (AR3)

*General Comment:*

The revised manuscript is still very well written. The comments to improve slightly the structure and the presentation of the methodology have been very well addressed. The authors gave relevant responses to each comment and improved a few sections of the article accordingly. The downscaling method used to select the climate model ensembles and the valorisation of the climate outputs to study the drought indices are well explained and a lot clearer than in the previous version. I believe this paper well addresses its research question, with now a clear methodology and comprehensive analyses of the future characteristics of drought in Australia and recommend it for publication.

*Response:*

*Thank you for the time spent reviewing the article and for your constructive comments which have improved the manuscript.*

*Specific Comment:*

l.179 and 185: Some « SPI/SPEI » left.

*Response:*

*We have fixed these two instances to 'SPI or SPEI' as suggested.*

*Specific Comment:*

L85-86: Issue in the sentence: "The downscaling was performed using dynamically downscaled using the Conformal Cubic Atmospheric Model (CCAM)"

*Response:*

We have fixed this sentence to:

*'The downscaling was performed using the Conformal Cubic Atmospheric Model (CCAM).'*